# Diffusion-based skin disease data augmentation with fine-grained detail preservation and interpolation for data diversity

**Mujung Kim**[1], **Jisang Yoo**[1]*, **Soonchul Kwon**[2]*, **Byung Jun Kim**[3], **Changsik John Pak**[4], **Chong Hyun Won**[5], **Suk-Ho Moon**[6], **Woo Jin Song**[7], **Han Gyu Cha**[8], **Kyung Hee Park**[9]

**1** Department of Electronic Engineering, Kwangwoon University, Seoul, Republic of Korea, **2** Graduate School of Smart Convergence, Kwangwoon University, Seoul, Republic of Korea, **3** Department of Reconstructive and Plastic Surgery, Seoul National University Hospital, Seoul, Republic of Korea, **4** Department of Plastic and Reconstructive Surgery, Asan Medical Center, Seoul, Republic of Korea, **5** Department of Dermatology, Asan Medical Center, Seoul, Republic of Korea, **6** Department of Plastic and Reconstructive Surgery, Seoul St. Mary's Hospital, Seoul, Republic of Korea, **7** Department of Plastic and Reconstructive Surgery, Soonchunhyang University Hospital, Seoul, Republic of Korea, **8** Department of Plastic Surgery and Rehabilitation Medicine, Soonchunhyang University Hospital, Bucheon, Republic of Korea, **9** Department of Nursing Science, The University of Suwon, Hwaseong, Republic of Korea

* jsyoo@kw.ac.kr (JY); ksc0226@kw.ac.kr (SK)

**Data availability statement:** All image files are available from the HAM10000 database

## Abstract

We propose a data augmentation technique utilizing a Diffusion-based generative deep learning model to address the issue of data scarcity in skin disease diagnosis research. Specifically, we enhanced the Stable Diffusion model, a Latent Diffusion Model (LDM), to generate high-quality synthetic images. To mitigate detail loss in existing Diffusion models, we incorporated lesion area masks and improved the encoder and decoder structures of the LDM. Multi-level embeddings were applied using a CLIP encoder-based image encoder to capture detailed representations, ranging from textures to overall shapes. Additionally, we employed pre-trained segmentation and inpainting models to generate normal skin regions and used interpolation techniques to synthesize synthetic images with gradually varying visual characteristics, while having limitations for clinical use, this approach contributes to enhanced data diversity and can be used as reference material. To validate our method, we conducted classification experiments on seven skin diseases using datasets combining synthetic and real images. The results showed improvements in classification performance, demonstrating the effectiveness of the proposed technique in addressing medical data scarcity and enhancing diagnostic accuracy.

## Introduction

In recent years, with the advancement of deep learning-based image processing technology, the potential applications of this technology have gained attention across various fields.

(https://dataverse.harvard.edu/dataset.xhtml?persistentId=doi:10.7910/DVN/DBW86T) and the PAD-UFES-20 database (https://data.mendeley.com/datasets/zr7vgbcyr2/1). The code for our diffusion model is available at https://github.com/raddshing/skin-disease-diffusion.

**Funding:** This research was supported by the MSIT (Ministry of Science and ICT), Korea, under the ITRC (Information Technology Research Center) support program (IITP-2025-RS-2023-00258639) supervised by the IITP (Institute for Information & Communications Technology Planning & Evaluation). And, the present research has been conducted by the Research Grant of Kwangwoon University in 2024. There was no additional external funding received for this study. The funders had no role in study design, data collection and analysis, decision to publish, or preparation of the manuscript.

**Competing interests:** The authors have declared that no competing interests exist.

In the medical field, attempts to utilize deep learning technology for diagnosis, education, and research are actively being made, with particular focus on research applying deep learning models to tasks such as diagnosis [1–3], detection [4–8], and segmentation [9–11].

Skin diseases represent a significant global health burden, affecting 30-70% of individuals worldwide [12]. According to the Global Burden of Disease Study 2013, skin conditions were the 4th leading cause of nonfatal disease burden globally [13]. In particular, skin cancer and pigmented lesions are important skin diseases where early diagnosis critically impacts patient prognosis. For melanoma, the 5-year survival rate reaches 99% when detected early, but decreases dramatically when metastasis occurs. Despite this substantial disease burden, dermatological diagnosis faces critical challenges. Many regions experience severe shortages of dermatologists, and accurate diagnosis of skin lesions requires specialized dermoscopy skills and clinical experience. Distinguishing between benign pigmented lesions and malignant melanoma is particularly challenging even for experienced specialists, further emphasizing the need for diagnostic support tools.

Deep learning models in the medical field require accurate and effective training, as even slight differences in performance can lead to severe consequences in real-world outcomes. To ensure such accuracy and effectiveness, it is essential to secure high-quality data that adequately reflects disease characteristics. However, collecting medical data poses several challenges [14], such as patient privacy concerns, stringent clinical data collection protocols, and the limited availability of data for rare diseases. These data scarcity issues are particularly acute in skin lesion diagnosis, where even public datasets like HAM10000 [15] have limited samples for certain lesion types.

To address these limitations, generative deep learning models, including GANs [16] and Diffusion models [17], have been explored for generating high-quality synthetic medical data. Diffusion models, in particular, are gaining attention for their stability in training and ability to generate diverse, high-quality images. Compared to GANs, they demonstrate superior performance in various tasks such as image generation [18,19], super-resolution [20,21], editing [22,23] and style transfer [24,25] while maintaining training stability. Recently, there has been active research in medical data generation [26–29] using Diffusion-based models such as Stable Diffusion [18] and Imagen [19].

Previous works have demonstrated the potential of generative models for medical data synthesis. For example, [30] proposed a two-stage image synthesis method based on ST-GAN [31] to generate diverse skin lesion images for both majority and minority classes from imbalanced skin lesion data. Derm-T2IM [32] developed a skin lesion image-text model based on the Stable Diffusion utilizing text prompts, demonstrating that stable model tuning is possible with only 1,400 images per class. Akrout et al [33] proved that synthetically generated images using Diffusion-based models showed similar skin disease classification performance compared to using real data. Additionally, Medfusion [34] demonstrated the potential of medical image synthesis by achieving superior performance in metrics such as FID, Precision, and Recall compared to GAN-based generative models through channel expansion of the Diffusion autoencoder.

However, existing models still face challenges, such as losing fine-grained details during image generation. This can be a critical drawback, especially in the medical domain where detailed characteristics such as disease patterns, textures, and shapes are crucial. Since even subtle differences in medical images can significantly impact actual diagnostic results. Addressing these limitations is essential for generating synthetic medical images that accurately represent disease-specific features and ensure their utility in clinical and diagnostic workflows.

In this study, we propose an improved method based on the Latent Diffusion Model(LDM) [18] to reduce the loss of detailed representations in medical data and generate high-quality synthetic data with improved diversity through latent space interpolation. Using the HAM10000 dataset [15], we generated synthetic images for seven types of skin diseases to address the challenge of creating detailed and diverse medical images. To incorporate additional feature information for enhancing detailed representations, we drew inspiration from ELITE [35], a study on personalized image generation tasks. Specifically, we utilized a pre-trained CLIP [36] image encoder to extract multi-level embeddings, capturing low-level features—such as disease patterns and textures—as well as high-level features, like overall disease morphology. These embeddings were integrated into the Diffusion U-net through separate adapter layers to strengthen detailed expression information. Additionally, to improve artifact reduction and generate realistic images, we adopted the autoencoder structural modifications proposed in Medfusion. By expanding the VAE [38] channel structure of Stable Diffusion, we enabled the model to learn richer image representations. Finally, to further enhance data variability, we applied latent space interpolation between synthesized lesion images and their corresponding inpainted normal counterparts. To achieve this, we utilized the segmentation masks of the synthesized samples and inpainting to extract corresponding normal region images. While the resulting interpolation is not fully reflective of clinically grounded severity concepts, it enables the generation of synthetic images with smoothly varying visual characteristics, thereby enriching training data and supporting reference-level diversity.

To validate the effectiveness of our proposed methodology, we performed classification downstream tasks combining synthetic and real data using classification models ranging from common ones such as VGG [40] and ResNet [41] to deep classifiers like ConvNext [42], Swin Transformer [43], EVA [44] and CoAtNet [45]. The experimental results showed an improvement in average classification accuracy from 80.92% to 84.15%, suggesting that synthetic data can alleviate the data scarcity problem and contribute to performance improvement.

While this study is similar to ELITE in utilizing multi-embeddings to improve detailed expressions in generated images, there are several key differences. While ELITE focused on personalized text-to-image generation task, our study aims at medical data augmentation and conducted experiments in the medical domain of skin disease data rather than in general domains. Additionally, while ELITE emphasized text-to-image learning by mapping detailed expression embeddings to text space through a separate mapping network, our study focused on visual learning using visual tokens.

Our contributions are as follows:

- To overcome the limitations of medical data with restricted data constraints, we generated high-quality synthetic data using a Diffusion-based model for seven classes in the skin disease domain for medical data augmentation. We validated the effectiveness of the synthesized skin data through classification downstream tasks.
- To compensate for the potential loss of detailed expressions in medical image generation, we introduced an improved Diffusion method that applies multi-level embeddings to Diffusion learning.
- We generated synthetic skin lesion images by utilizing the segmentation masks of the synthesized samples and inpainting to extract normal region images, followed by latent space interpolation between normal and synthesized lesions. This approach enabled the creation of synthetic images with smoothly varying visual characteristics that, while not fully aligned with clinically grounded severity concepts, can be utilized for reference purposes and to enhance data diversity.

## Related works

### Medical data augmentation

Data augmentation [46] is a crucial technique used to address overfitting problems and performance degradation that can occur during deep learning model training in situations with insufficient data. Traditional data augmentation techniques involve complementing training data by applying simple geometric transformations to the original data, such as rotation, rescaling, translation, flipping, and affine transformations. However, medical data, unlike general data, has complex and specialized characteristics, making it difficult to generate new data that sufficiently reflects the detailed characteristics of medical data through such simple transformations alone. Furthermore, obtaining original medical data itself is very challenging due to constraints such as data availability, privacy protection, and strict collection procedures.

To address these issues, specialized deep learning-based data augmentation techniques for the medical domain have been actively researched recently. Generative Adversarial Networks (GANs) [16] have gained attention for their ability to generate high-quality data and are being utilized in various medical data augmentation studies [48–50]. For example, in Zanini et al [47], DCGAN [51] was used to generate normal electromyography signals, and then Style Transfer [52] techniques were employed to generate virtual data reflecting the style of Parkinson's disease patients' EMG signal data. Additionally, in [48] addressing object detection and multi-organ localization problems in CT images, CycleGAN [53] was used for data augmentation and combined with the YOLO [54] model to improve detection efficiency. Furthermore, MG-GAN [55], a GAN model designed to generate data following Gaussian distribution, demonstrated improvements in data augmentation and classification performance for gene expression datasets. Recently, Diffusion models have gained attention for their ability to generate high-quality images through stable learning methods, and research on medical data augmentation using these models [56–58] is actively being conducted. Kai Packhauser et al [59]. proposed a Latent Diffusion Model (LDM) [18] with privacy-enhancing techniques for chest X-ray data, presenting an approach to overcome data collection constraints by generating class-conditional images. Furthermore, [60] used DALLE2 [61] to generate synthetic data to address the generalization problem of underrepresented skin type groups in the Fitzpatrick 17k dataset [62], demonstrating improvements in skin disease classification performance. In Medfusion [34], the VAE channel structure of the Stable Diffusion [18] model was expanded to synthesize medical images including chest X-rays, iris, and pathology data, showing superior performance in quantitative metrics such as FID, Precision, and Recall compared to GAN-based models. Likewise, Wang *et al.* [63] leveraged a Variational Autoencoder to synthesize biochemical-blood-test profiles for stage II/III pancreatic-cancer patients, cutting the survival-prediction mean absolute error to roughly 10 days and underscoring the clinical value of generative augmentation in precision-medicine settings.

In this study, we conducted our study using Stable Diffusion, a Diffusion-based model showing outstanding performance in the medical domain, for generating synthetic images of skin diseases.

### Diffusion probabilistic models

Diffusion models [17,64] are deep learning generative models based on the principles of the diffusion process, which generate high-quality data by gradually adding noise to data and reversing this process for restoration. Diffusion Probabilistic Models (DPM) [64] introduced the principle of learning data distribution from simple probabilistic distributions using

Markov chains. DPM generates data through a probabilistic process that gradually adds noise to data in the forward process and restores data in the reverse process.

Denoising Diffusion Probabilistic Models (DDPM) [17] is an advanced model based on the principles of DPM, specialized in image generation tasks. DDPM simplified the learning process by introducing a method to predict noise at each step instead of restoring data. It performs denoising gradually and efficiently restores data using a Markov chain composed of T steps. DDPM shows higher training stability compared to other generative models like GANs [16] and can effectively model complex data distributions. However, DDPM has a limitation in its slow sampling speed.

To address this limitation, Denoising Diffusion Implicit Models (DDIM) [65] improved sampling speed by transforming DDPM's reverse sampling process into a deterministic approach. Instead of probabilistic sampling, it uses a deterministic sampling method that produces consistent results under specific conditions, significantly reducing the number of sampling steps. This enabled efficient high-quality data generation.

Score-Based Generative Modeling through SDE [66] modeled the diffusion process in a more generalized mathematical framework by interpreting the data transformation process as a Stochastic Differential Equation (SDE). This expanded its applicability to various generative tasks such as super-resolution, image restoration, and style transfer. SDE provides a more flexible and generalized approach beyond diffusion models and has demonstrated powerful performance even in complex data generation problems.

Latent Diffusion Models (LDM) [18] were proposed to address the high computational cost and memory usage problems associated with DDPM-based models performing direct computations in pixel space when generating high-resolution images. LDM compresses data into a latent space before performing the diffusion process in this space. For this, it uses an autoencoder like VAE [38] to transform data into latent space and restore it back to original data through a reconstruction process. This approach significantly reduced computational and memory requirements while showing effective performance even in processing high-resolution images and complex data. However, LDM heavily depends on the performance of the autoencoder and may lose some data information during the compression and restoration process to and from latent space.

In this study, inspired by existing LDM-based studies for high-resolution medical image data generation, we conducted experiments on VAE structure modification suitable for medical image data and methods to reduce detail loss, based on Stable Diffusion, an LDM model.

## Severity-based medical data

While data augmentation techniques have been widely studied, research on severity-based medical image generation remains relatively limited. Existing studies have primarily focused on severity classification tasks [67–69], training models to predict disease severity using pre-existing datasets. These studies have demonstrated the importance of severity-based data in clinical decision-making but have not addressed the challenge of generating new images that reflect various levels of severity.

The Uncertainty-Guided Diffusion Models (UGDM) [70] utilized pre-trained classifier guidance during the diffusion generation process to generate data near class boundaries based on entropy and margin probability uncertainty metrics, demonstrating the potential to control disease severity. Similarly, Takezaki et al. [71] proposed the Ordinal Diffusion Model (ODM), focusing on generating medical images for classes with ordinal relationships (e.g., severity levels) in the retinal endoscopy domain. ODM introduced methods to control ordinal relationships during the image generation process.

Despite these efforts, research specifically addressing image generation based on severity remains limited. To complement this gap, this paper applies existing techniques, such as segmentation masks and inpainting, to design a method for generating synthetic images through latent space interpolation between normal and lesion states. This approach produces images with smoothly varying visual characteristics that, while not fully aligned with clinically grounded severity concepts, can serve as reference samples and contribute to enhanced data diversity in visual characteristics of generated skin lesion images.

## Materials and methods

### Overview

This study proposes a novel framework for Diffusion-based data augmentation that preserves fine details, reduces artifacts in generated images, and enhances visual diversity through latent space interpolation latent space interpolation. Fig 1 illustrates the overall workflow of the proposed model, highlighting its key components and procedures.

We performed pre-training by expanding the existing 4-channel VAE [38] of the Stable Diffusion model [18] to 8 channels. This enhanced the representational power of the latent representation space and effectively reduced structural degradation and artifacts that could occur during image generation.

Prior to training, the model input data is preprocessed using lesion masks. This process extracts skin lesion areas and removes unnecessary backgrounds, designed to allow the model to focus more on the key features of lesions. Extracted lesion images are transformed into multi-level embeddings using the CLIP [36] image encoder. The generated embeddings contain detailed features and overall structural information of lesions and are passed to Transformer blocks within the Diffusion U-Net. These blocks effectively incorporate detailed lesion characteristics into model learning through specially designed Adapter Layers.

In the image generation process, latent space interpolation is utilized to generate synthetic samples with gradually varying visual characteristics. To achieve this, normal images with the same context as the normal regions of the generated skin disease samples are created using pre-trained segmentation [72] and inpainting models [73]. These generated normal images and lesion images are then used together to generate data corresponding to a continuum of visual transitions between normal and lesion appearances.

This design goes beyond simple image synthesis and contributes to providing data that is useful for diagnosis and analysis.

Each component is explained in detail in the following sections.

### Datasets

In this study, we utilized the HAM10000 [15] dataset for training and evaluating our Diffusion-based data augmentation model. HAM10000 dataset is a comprehensive collection of dermoscopy images built to support automated classification of skin lesions, providing a total of 10,015 images covering seven major skin disease classes. The class composition of the dataset is as follows:

- **Actinic keratoses/intraepidermal carcinoma (AKIEC)** — 327 images.
  - Clinical: Non-invasive variants of squamous cell carcinoma that can be treated locally without surgery. More common on sun-exposed areas, with actinic keratoses typically on the face and Bowen's disease on other body sites.
  - Morphological: Commonly show surface scaling and are often devoid of pigment. May progress to invasive squamous cell carcinoma.

*Step1: Pretrain VAE*

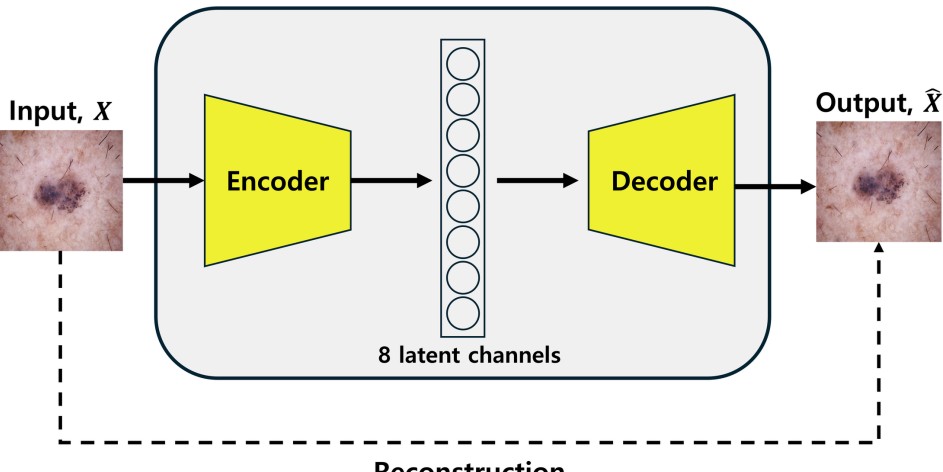

*Step2: Diffusion Process with multi-embeddings*

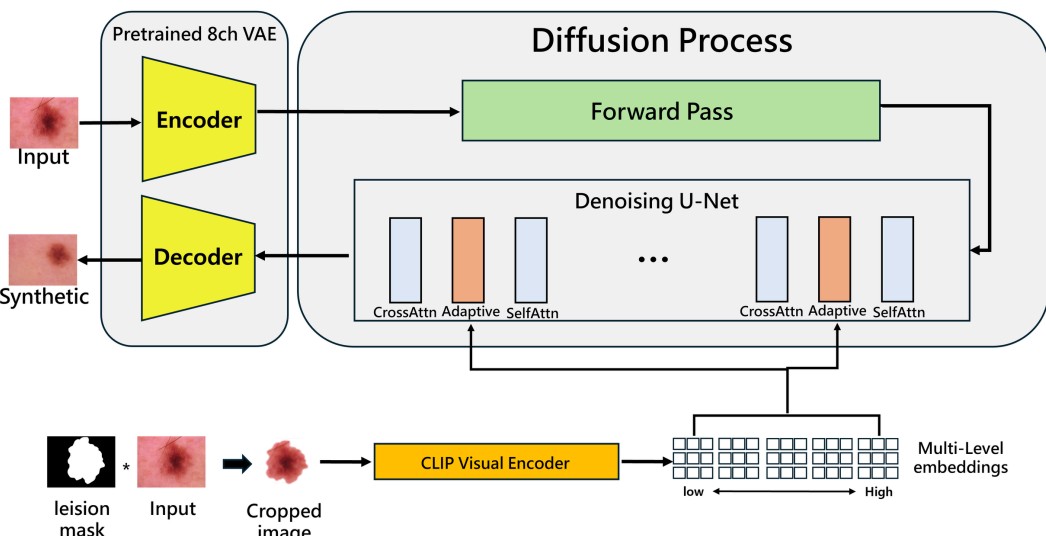

**Fig 1. The workflow of our method.** In Stage 1, the VAE latent channels in the existing Stable Diffusion model are expanded from 4 to 8, and the VAE is pre-trained to enhance representational capacity. In Stage 2, the pre-trained 8-channel VAE is used to configure the input and output layers. For detailed representation learning, a pre-trained CLIP image encoder extracts multi-level embeddings at five scales. These embeddings are injected into the denoising UNet during the diffusion process through additional adapter layers, leading to the generation of the final synthesized images.

- **Basal cell carcinoma (BCC)** — 514 images.
    - Clinical: Common epithelial skin cancer that rarely metastasizes but grows destructively if untreated.
    - Morphological: Appears in different variants (flat, nodular, pigmented, cystic) with varying dermatoscopic presentations.
- **Benign keratosis-like lesions (BKL)** — 1,099 images.
    - Clinical: Generic class including seborrheic keratoses ("senile wart"), solar lentigines, and lichen-planus like keratoses (LPLK).

– Morphological: Show inflammation and regression patterns. Particularly challenging dermatoscopically as they can mimic melanoma features, often requiring biopsy for diagnostic certainty.

- **Dermatofibroma (DF)** — 115 images.
  - Clinical: Benign skin lesion considered either a benign proliferation or inflammatory reaction to minimal trauma.
  - Morphological: Most commonly presents with reticular lines at the periphery and a central white patch denoting fibrosis.
- **Melanoma (MEL)** — 1,113 images.
  - Clinical: Malignant neoplasm derived from melanocytes. Can be cured by simple surgical excision if detected early. Includes all variants including melanoma in situ.
  - Morphological: Usually chaotic in appearance with asymmetric distribution of colors and structures. Specific criteria depend on anatomic site.
- **Melanocytic nevi (NV)** — 6,705 images.
  - Clinical: Benign neoplasms of melanocytes appearing in multiple variants.
  - Morphological: In contrast to melanoma, usually symmetric with regard to distribution of color and structure. Variants may differ significantly from a dermatoscopic perspective.
- **Vascular lesions (VASC)** — 142 images.
  - Clinical: Range from cherry angiomas to angiokeratomas and pyogenic granulomas. Include hemorrhage in this category.
  - Morphological: Dermatoscopically characterized by red or purple color and solid, well-circumscribed structures known as red clods or lacunes.

An example of the HAM10000 dataset is shown in Fig 2. Each image is provided with metadata including patient age, gender, anatomical location of the lesion, and lesion size. HAM10000 is a representative dataset widely used in academia for training and evaluating machine learning models for skin disease classification. During the experiment, the dataset was split into training, validation, and test sets, and a weight-based data balancing technique was applied to mitigate the class imbalance problem between classes. All images were preprocessed identically in terms of size and quality to maintain consistency in the training process. All images were resized to 256 × 256 pixels, and lesion masks were used to remove unnecessary backgrounds and focus on the main lesion areas. This enables the model better learn the detailed characteristics of lesions. For CLIP-based multi-level embedding extraction, the same images were resized to 224 × 224 pixels using bicubic interpolation and then normalized using the pre-trained CLIP model's normalization parameters to ensure optimal feature extraction. In all image preprocessing steps, normalization was applied to standardize pixel intensity distribution across the dataset. Additionally, following standard practice, the dataset was split into training, validation, and test sets, and weight-based data balancing was applied to mitigate the class imbalance problem between classes.

## Enhanced Variational Autoencoder (8-channel VAE)

The Stable Diffusion models [18] perform the diffusion process in latent space using a Variational Autoencoder (VAE) [38] rather than working directly in the high-dimensional data space of images. This latent space represents compressed image data, and the performance of the VAE directly impacts the generation quality of Latent Diffusion Models (LDM). While the standard 4-channel VAE architecture is widely used in LDMs, it often causes loss of detailed

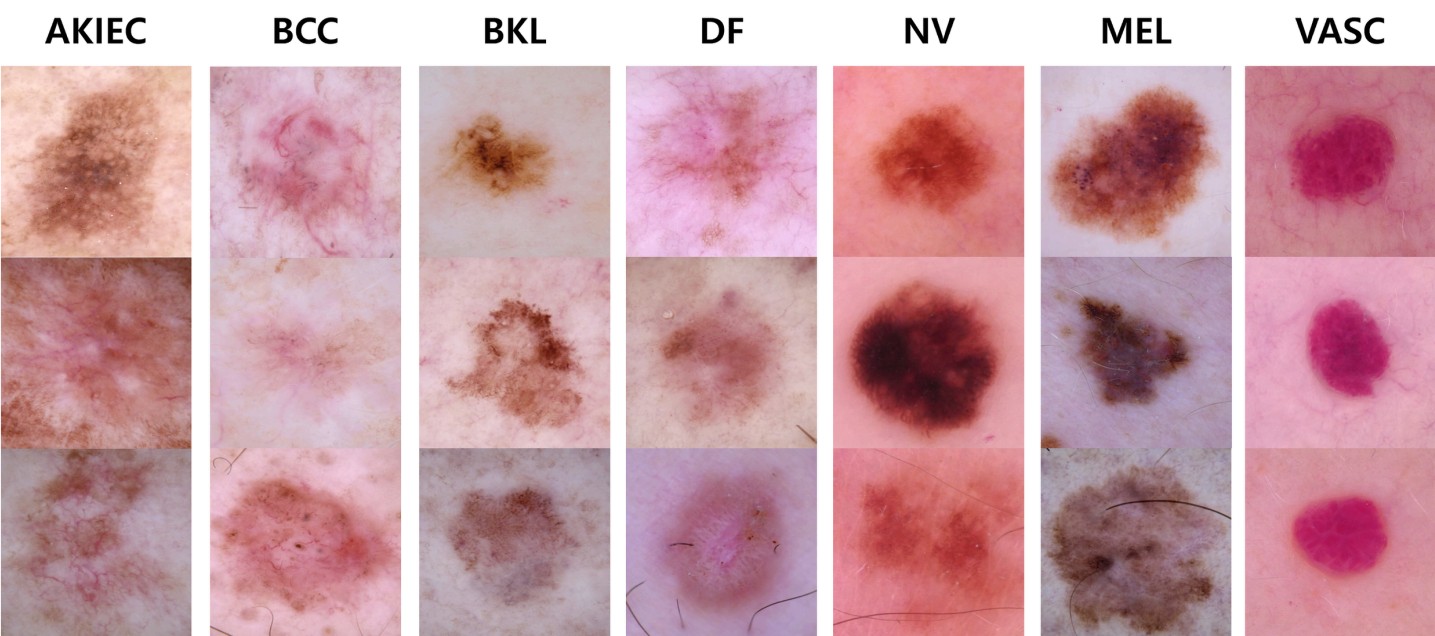

| AKIEC | BCC | BKL | DF | NV | MEL | VASC |

**Fig 2. An example of the HAM10000 dataset.** Consisting of 10,015 images across seven skin tumor classes.

representations and generates artifacts when processing medical images. These issues are particularly critical in the medical imaging domain, where fine-grained textures and patterns play a crucial role.

Medfusion [34] introduced the 8-channel VAE to expand latent space representation for medical imaging tasks across fundus, chest X-rays, and pathological image. Inspired by Medfusion, we applied the 8-channel VAE to the HAM10000 [15] dataset to address challenges in the skin lesion domain. Skin lesion data requires accurate reconstruction of detailed textures, patterns, and shapes, which are essential for capturing disease-specific characteristics. To address this, we implemented an 8-channel VAE that doubles the latent space capacity compared to the standard 4-channel VAE. While this expansion leads to a slightly reduced compression rate, it significantly improves the quality of reconstructed images and reduces artifact generation.

The loss functions used during training are as follows:

$$\mathcal{L}_{\text{VAE}} = \lambda_{\text{rec}}\big(\mathcal{L}_1(\mathbf{x}, \hat{\mathbf{x}}) + \text{SSIM}(\mathbf{x}, \hat{\mathbf{x}})\big) + \lambda_{\text{KL}}D_{\text{KL}}\big(q_\phi(\mathbf{z}|\mathbf{x})\|p(\mathbf{z})\big) + \lambda_{\text{perc}}\text{LPIPS}(\mathbf{x}, \hat{\mathbf{x}}) \qquad (1)$$

The loss function of this VAE model consists of Reconstruction Loss, Latent Space Regularization Loss, and Perceptual Loss. First, the reconstruction loss focuses on minimizing the difference between the original and reconstructed images. For this purpose, it uses L1 loss (absolute value loss) to calculate pixel-wise differences and employs Structural Similarity Loss(SSIM Loss) [74] to maintain structural similarity between the original and reconstructed images. SSIM Loss measures the structural differences between the original and reconstructed images, enabling more effective reconstruction of lesion details. Next, the latent space regularization loss uses KL Divergence Loss to normalize the latent space distribution. This loss guides the model to learn stable and generalized representations in the latent

space. Lastly, perceptual loss uses LPIPS [75] (Learned Perceptual Image Patch Similarity) to evaluate perceptual similarity of images and induce high-quality reconstruction.

The pre-trained 8-channel VAE is used at the input and output stages of the Stable Diffusion model, with the VAE weights remaining fixed during the diffusion training process. This approach demonstrated that while minimizing the VAE's impact on the quality and detailed representation of generated images, it can support stable and high-quality data generationwith reduced artifacts.

## Preserving visual details through multi-level embeddings

Latent Diffusion Models (LDM) [18] encounter the challenge of losing the preservation of fine-grained details during image compression and reconstruction in latent space. This loss of detail can be especially concerning in synthetic data generation using medical images, as it may lead to severe consequences. To address this issue, our study designed a method to preserve and effectively integrate the fine-grained visual details of original data in the skin disease domain into the diffusion training process.

First, to eliminate unnecessary background regions and focus on the characteristics of the lesion area, we generate images containing only the lesion region using lesion segmentation masks. The lesion extraction image is defined as follows:

$$\mathbf{x}_l := \mathbf{x}_i \cdot \mathbf{m}_i, \tag{2}$$

where $x_i$ is the input image, $m_i$ is the lesion mask image of the corresponding image, and $x_l$ is the extracted lesion image.

The extracted lesion images are then processed through a pre-trained CLIP [36] image encoder to generate embeddings that reflect visual details. Specifically, we employ the ViT-H/14 backbone from OpenCLIP [37] and freeze all of its parameters; the layer-specific two-layer MLPs are the only learnable components and are trained jointly with the diffusion UNet.

We extract hidden representations from transformer blocks at layers $\{5, 11, 17, 23, 31\}$. The input images are first resized to $224 \times 224$ using bicubic interpolation and normalized using OpenCLIP's standard normalization parameters ($\mu = [0.48, 0.46, 0.41]$, $\sigma = [0.27, 0.26, 0.28]$).

For each selected layer $l$, we extract both the CLS token and the top-$k$ patch tokens based on their L2 norm:

$$\begin{aligned} \mathbf{h}^{(l)} &= [\mathbf{h}_{\text{CLS}}^{(l)}, \mathbf{h}_{\text{patch}}^{(l)}], \\ \mathbf{h}_{\text{patch}}^{(l)} &= \text{top-}k(\{\mathbf{p}_i^{(l)}\}_{i=1}^N, \|\mathbf{p}_i^{(l)}\|_2) \end{aligned} \tag{3}$$

where $N$ is the number of patch tokens, and we select $k = 32$ patches with the highest L2 norms to capture the most salient visual features.

Each layer's extracted tokens (concatenated CLS + top-32 patches, resulting in 33 tokens per layer) are then projected through a two-layer MLP with SiLU activation:

$$\mathbf{e}_l = \text{MLP}_l(\mathbf{h}^{(l)}) = W_2^{(l)} \cdot \text{SiLU}(W_1^{(l)} \cdot \mathbf{h}^{(l)}) \tag{4}$$

where $W_1^{(l)} \in \mathbb{R}^{d_{\text{CLIP}} \times 1024}$ and $W_2^{(l)} \in \mathbb{R}^{1024 \times 1024}$ project the CLIP features to a unified dimension $d_{\text{proj}} = 1024$.

By extracting features from five different transformer layers, we capture a hierarchy of visual representations: early layers (e.g., layer 5) capture low-level features such as textures and color patterns, intermediate layers (e.g., layers 11, 17) capture local structures and patterns, while deeper layers (e.g., layers 23, 31) capture high-level semantic features representing

overall lesion morphology and structure. Fig 3 visualizes the feature representations from all extracted layers using PCA projection to RGB space, demonstrating the progressive abstraction of visual information across the transformer layers. In these visualizations, pronounced color differences between adjacent patches indicate that the model distinguishes different regions such as lesion boundaries and detailed morphological structures. Notably, as layer depth increases, the inter-patch color variations become more prominent, confirming that our multi-level extraction approach effectively captures fine-grained visual details essential for accurate lesion representation.

During the diffusion training process, visual tokens are extracted from the raw RGB images (before latent encoding) and injected into the U-Net encoder through lightweight adapter modules. The U-Net architecture consists of 4 resolution levels with hidden dimensions $\{256, 256, 512, 1024\}$. For each resolution level $i$ in the U-Net encoder, the visual tokens

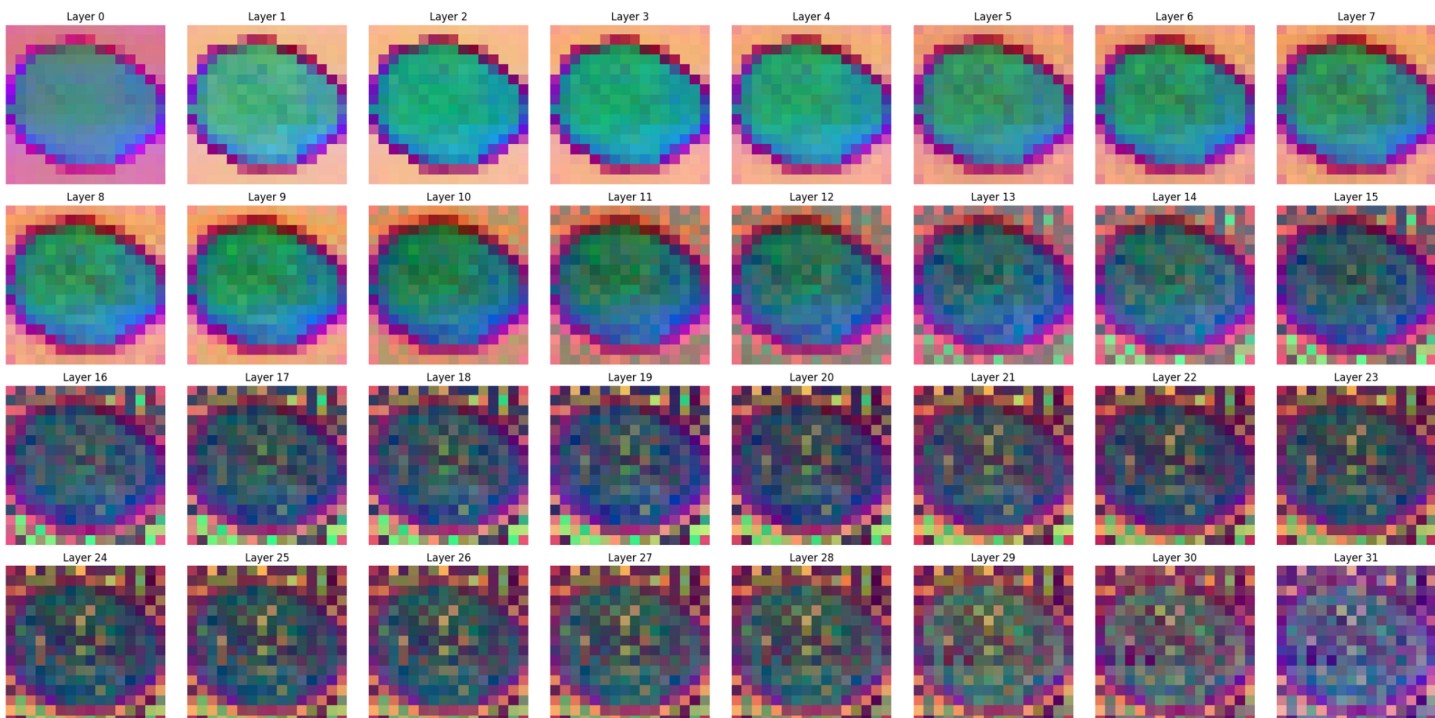

**Fig 3. Multi-level feature visualization of skin lesion images.** PCA projection of features extracted from all 32 layers (0-31) of the OpenCLIP ViT-H/14 visual encoder into RGB space. Each layer's feature map is projected using the first three principal components mapped to RGB channels. The pronounced color variations between adjacent patches indicate the model's ability to distinguish different regions such as lesion boundaries and morphological structures, with deeper layers showing more prominent inter-patch differences for fine-grained detail representation.

from the corresponding CLIP layer are first projected to match the channel dimension:

$$\mathbf{v}_i = \text{MLP}_i(\mathbf{e}_l) = W_2^{(i)} \cdot \text{SiLU}(W_1^{(i)} \cdot \mathbf{e}_l) \tag{5}$$

where the MLP projects from $d_{\text{proj}} = 1024$ to the hidden dimension $d_i \in \{256, 256, 512, 1024\}$ of the U-Net level.

Inspired by GLIGEN [77], we employ adapter modules that perform cross-attention between U-Net features and visual tokens. The adapter module modulates the U-Net features as follows:

$$\mathbf{h}' = \mathbf{h} + \beta \cdot \text{Adapter}(\mathbf{h}, \mathbf{v}_i) \tag{6}$$

where $\mathbf{h} \in \mathbb{R}^{B \times HW \times d_i}$ represents the flattened spatial features from the U-Net encoder block, $\beta = 0.1$ is a scaling factor, and the adapter performs cross-attention with 8 attention heads. To maintain computational efficiency, adapters are only applied to feature maps where the spatial resolution satisfies $HW \leq 4096$, and are specifically excluded from downsampling blocks.

Unlike ELITE which maps CLIP features to the text embedding space and modifies cross-attention mechanisms, our approach directly injects visual information into the U-Net's visual feature stream through these lightweight adapters.

## Latent space interpolation-based image generation

In this study, we propose a novel approach based on segmentation and inpainting to generate a pair of disease and corresponding pseudo-normal images, followed by latent space interpolation to produce images with smoothly varying visual characteristics.

The process is illustrated in Fig 4. The method first produces a pair of disease/pseudo-normal images and then performs latent interpolation.

First, sample images for each class are generated using a pre-trained Latent Diffusion Model (LDM). These generated sample images are defined as $x_{\text{disease}}$

Next, a segmentation model, MFSNet [72], is trained on the HAM10000 dataset to segment the lesion regions of $x_{\text{disease}}$ and generate a mask, $x_m$. The lesion mask is expressed as follows:

$$\mathbf{x}_m = \text{MFSNet}(\mathbf{x}_{\text{disease}}), \tag{7}$$

Subsequently, an inpainting model, Large Mask Inpainting (LaMa) [73], is trained on the HAM10000 dataset. Using the lesion mask, the model removes the lesion regions and inpaint the normal regions by considering the surrounding skin context. The generated normal image is defined as $x_{\text{normal}}$, which can be expressed as follows:

$$\mathbf{x}_{\text{normal}} = \text{LaMa}(\mathbf{x}_{\text{disease}} \odot (1 - \mathbf{x}_m)). \tag{8}$$

To enable interpolation in the latent space, both normal and diseased images are mapped to the latent space using latent space inversion. A pre-trained 8-channel VAE encoder is used to encode $x_{\text{normal}}$ and $x_{\text{disease}}$ into their respective latent representations:

$$\mathbf{z}_{\text{normal}} = \mathcal{E}(\mathbf{x}_{\text{normal}}), \quad \mathbf{z}_{\text{disease}} = \mathcal{E}(\mathbf{x}_{\text{disease}}), \tag{9}$$

where $z_{\text{normal}}$ and $z_{\text{disease}}$ represent the latent embeddings of the normal and diseased images, respectively.

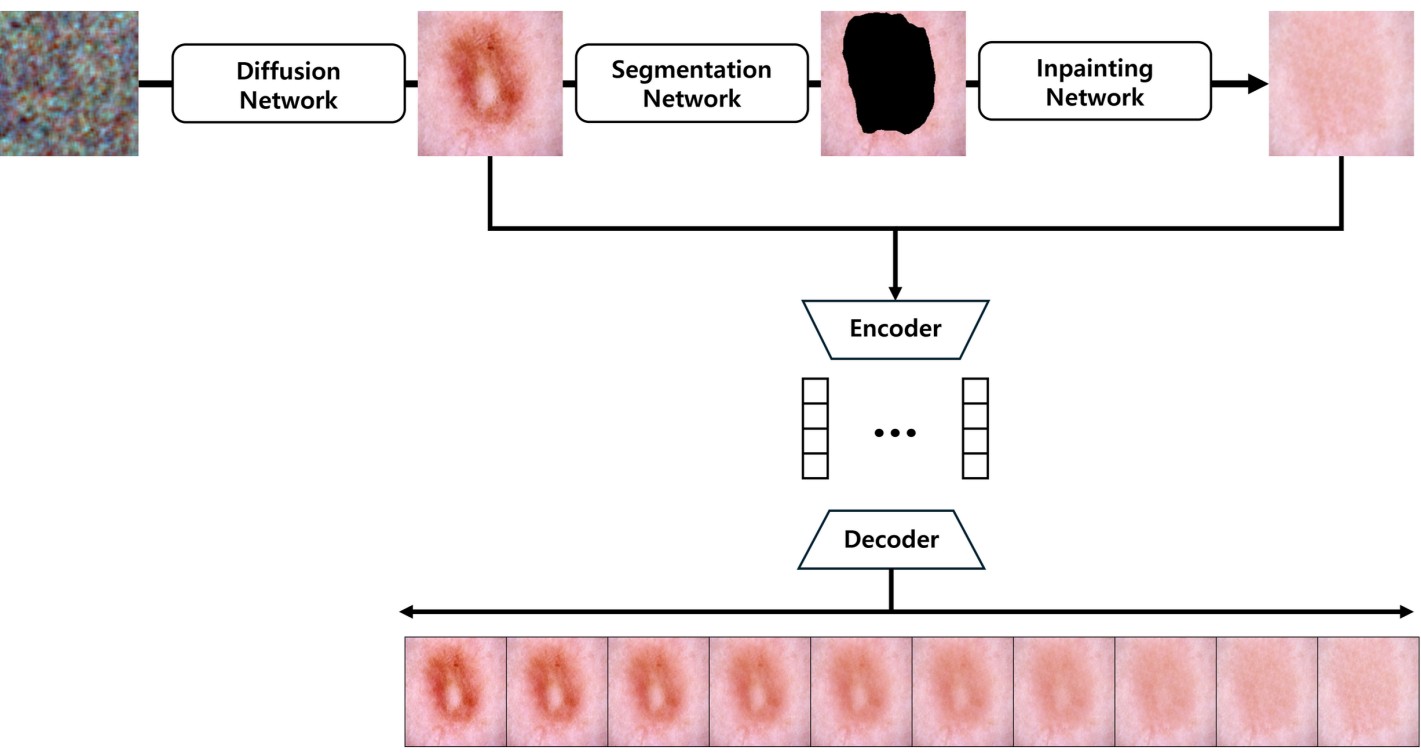

**Fig 4. Latent space interpolation-based sampling workflow.** Disease images are generated from noise using a diffusion model and then segmented to isolate lesion regions. These regions are inpainted to obtain pseudo-normal counterparts while preserving skin context. Both disease and pseudo-normal images are mapped to the latent space via a pre-trained VAE. Latent interpolation is then performed between the two extremes. While this produces visually smooth transitions, the interpolation axis reflects a heuristic binary lesion–normal direction rather than a clinically grounded severity concept.

Latent space interpolation is performed by linearly combining the latent representations of the two states (normal and diseased):

$$\mathbf{z}_{\text{interpolated}} = (1 - \alpha)\mathbf{z}_{\text{normal}} + \alpha\mathbf{z}_{\text{disease}}, \quad \alpha \in [0, 1], \tag{10}$$

where $\alpha$ is the interpolation coefficient, which gradually increases from 0 to 1 to represent a smooth transition from the normal to the leision.

Finally, the interpolated latent representations are decoded back to the image space using the VAE decoder of the diffusion model:

$$\mathbf{x}_{\text{interpolated}} = \mathcal{D}(\mathbf{z}_{\text{interpolated}}). \tag{11}$$

Through this process, we generate synthetic images with gradually varying visual characteristics, starting from the normal image. This approach combines segmentation, inpainting, and latent space interpolation to represent continuous transitions between normal and diseased states. While effective for data augmentation, the interpolation rests on a binary normal–lesion axis, so the generated sequence should be regarded as a heuristic visual progression rather than a rigorously calibrated severity ladder.

## Implementation details

Stable Diffusion v1-4 served as the base model, but several components were modified to align with the objectives of this study. The original CLIP text encoder was removed, and conditioning is now provided solely through class labels. Specifically, we employ a learnable embedding layer that maps class labels (0: AKIEC, 1: BCC, 2: BKL, 3: DF, 4:NV, 5: MEL, 6:VASC) to 1024-dimensional embedding vectors, which are then integrated into the model through the retained cross-attention blocks. This class-conditional approach enables the model to generate lesion-specific features while maintaining the architectural benefits of the original Stable Diffusion framework. To ensure a fair comparison, both the baseline and proposed methods underwent identical architectural modifications: the same LabelEmbedder for class conditioning, identical UNet configuration (8-channel input/output with hidden channels [256, 256, 512, 1024]), and matching training hyperparameters. The VAE was pre-trained with the Adam optimizer to encode $256 \times 256$ images into a $32 \times 32$ latent space (downsampling factor 8), using a weighted sum of KL, L1, L2 and SSIM losses (batch 16).Two latent-width variants are compared: the default 4-channel setting and an 8-channel setting that doubles capacity while keeping the spatial size unchanged. The baseline model maintains the original Stable Diffusion's 4-channel VAE configuration, while our proposed method employs the 8-channel variant to enhance representational capacity for medical imaging features.

The diffusion process consists of a forward stage that perturbs latents with Gaussian noise over 1 000 steps and a backward stage that denoises them. The UNet is trained with AdamW (batch 16, learning-rate $1 \times 10^{-4}$) under an L1 noise-prediction loss.

The noise scheduler employs a scaled linear beta schedule ($\beta_{\text{start}} = 0.002$, $\beta_{\text{end}} = 0.02$) across 1,000 timesteps, identical for both baseline and proposed methods.

During sampling, we employed class-specific strategies to optimize generation quality for each skin lesion type. Instead of using a uniform approach, we varied the sampling method (DDPM or DDIM), classifier-free guidance scale (3.0 or 5.0), and number of denoising steps (50-1000) based on empirical observations of each class's generation characteristics. For instance, melanoma (MEL) generation utilized DDPM with 1000 steps and cfg=5.0, while benign lesions like BCC employed faster DDIM sampling with 100 steps and cfg=3.0. This adaptive sampling strategy was determined experimentally to achieve optimal visual quality for each lesion type while balancing computational efficiency.

VAE training was conducted using both two NVIDIA RTX 3090 GPUs (24GB each) and Colab A100 GPU(40GB). The VAE was trained with a fixed learning rate of 1e-4, while the diffusion model employed a cosine annealing schedule with warm-up. All experiments were implemented in Python 3.9.19 with PyTorch 1.12 and CUDA 11.4 on a 2.9 GHz Intel Core i7-10700 processor.

## Results

### Reconstruction with 8-channel VAE

In this section, we conducted reconstruction quality experiments in the HAM10000 skin disease domain using the proposed 8-channel VAE approach. The 8-channel VAE was designed to expand the representational capacity of the latent space compared to the conventional 4-channel VAE, aiming to reduce artifacts. We applied this architecture to the HAM10000 dataset to evaluate the reconstruction quality of skin disease data.

We compared the performance of base model's 4-channel VAE and 8-channel VAE used in this study. both quantitatively and visually, demonstrating that the 8-channel VAE showed superior results on the HAM10000 dataset. Fig 5 shows the comparison of reconstructed

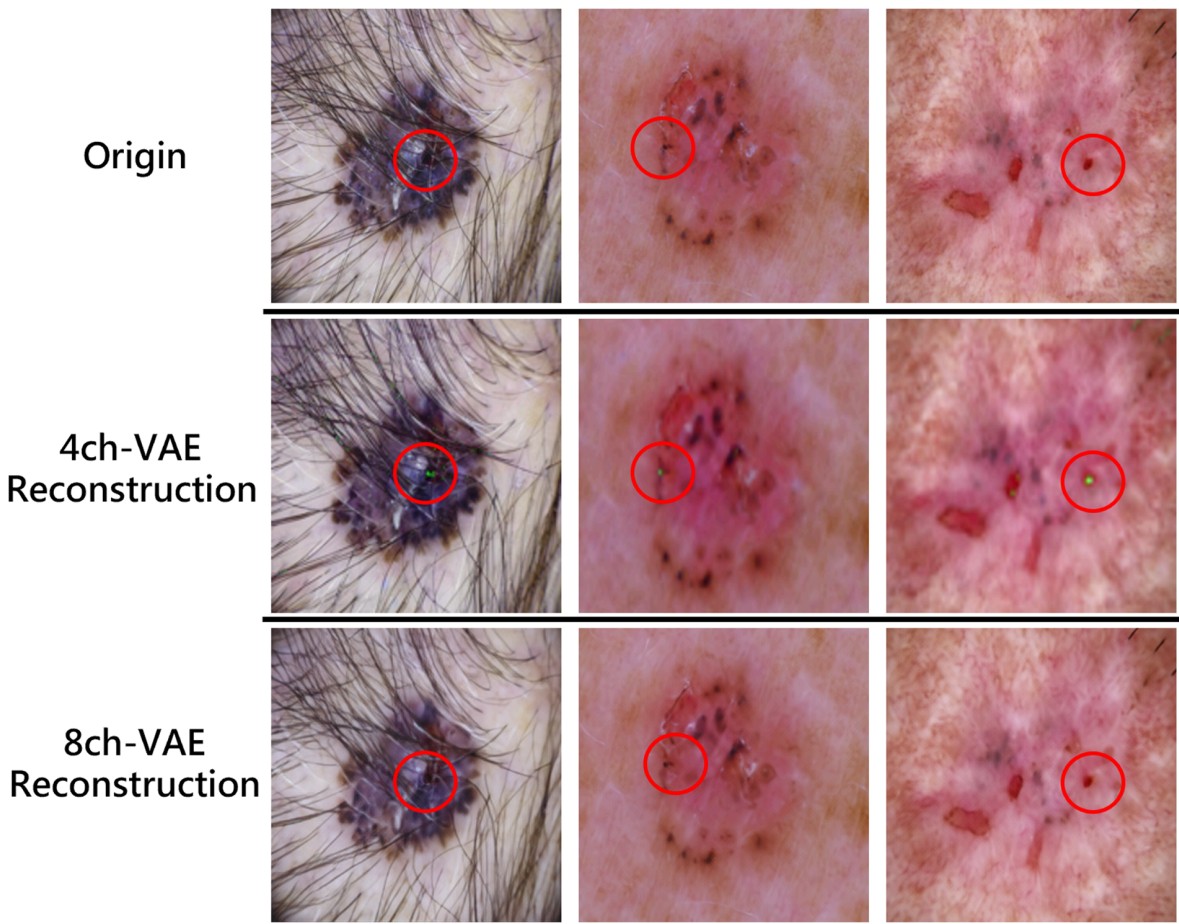

**Fig 5. Visual comparison of reconstruction results: 4-Channel VAE vs. 8-Channel VAE.** In the areas marked with red circles, the 4-channel VAE exhibits subtle artifacts, whereas the 8-channel VAE demonstrates improvements, mitigating these issues.

images generated by 4-channel VAE and 8-channel VAE. While images generated by the 4-channel VAE showed distorted artifacts around the lesions, the 8-channel VAE produced more natural results with improved artifacts. This suggests that the expanded latent space of the 8-channel VAE can prevent lesion information distortion in the skin domain.

Table 1 presents the quantitative analysis results for 4-channel and 8-channel VAE. The 8-channel VAE showed better performance across all evaluation metrics, including MSE, LPIPS Score [75], and MS-SSIM [76]. MSE (Mean Squared Error) measures the average squared difference between the reconstructed image and the original image at the pixel level. A lower MSE indicates that the reconstructed image is closer to the original. The 16% improvement in MSE achieved by the 8-channel VAE demonstrates a significant reduction in pixel-level loss, suggesting that the model can more accurately reproduce the overall structure of the lesion. LPIPS (Learned Perceptual Image Patch Similarity) evaluates the perceptual similarity between two images based on human vision. Lower LPIPS values indicate higher visual similarity. Although the improvement in LPIPS was marginal at approximately 3%, this still suggests that the 8-channel VAE can reproduce subtle textures and lesion shapes more naturally. MS-SSIM (Multi-Scale Structural Similarity) assesses the structural similarity of images across

**Table 1. Quantitative comparison between 4ch VAE vs 8ch VAE.** In the comparison of reconstructed image quality, the 8-channel VAE showed superior results across MSE, LPIPS, and MS-SSIM metrics.

|  | MSE($10^{-3}$)↓ | LPIPS↑ | MS-SSIM↑ |
|---|---|---|---|
| 4ch-VAE | 0.38 | 0.91 | 0.95 |
| 8ch-VAE | **0.21** | **0.94** | **0.97** |

multiple scales, focusing on details like texture and contrast. The 2% improvement in MS-SSIM indicates that the 8-channel VAE preserves fine lesion structures and surrounding skin context more effectively. This is particularly significant for medical images, where maintaining structural details plays a crucial role in accurate identification and diagnosis of lesions.

In conclusion, the 8-channel VAE outperformed the conventional 4-channel VAE in terms of pixel accuracy, perceptual similarity, and structural preservation. These results demonstrate that the 8-channel VAE effectively restores detailed representations of lesions while reducing artifacts. This improvement in image quality can enhance the reliability of diagnostic models, providing a significant contribution to medical imaging applications.

### Evaluation of synthetic image quality and fine detail preservation

We conducted visual and quantitative evaluations of synthetic images generated by the improved Diffusion model with multi-level embeddings for fine detail preservation.

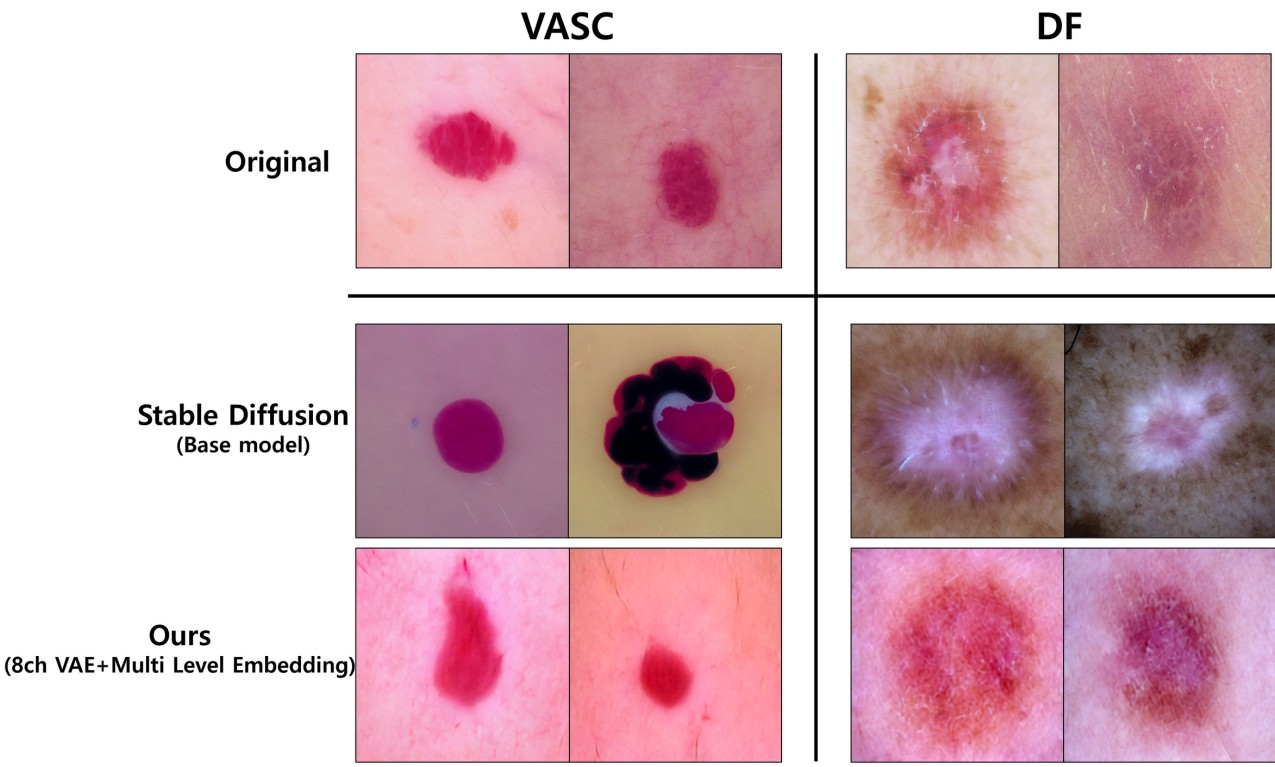

**Fig 6. Visual comparison between the baseline model and our method.** In the "VASC" class, which has relatively fewer training samples, the samples generated by Stable Diffusion (baseline model) fail to accurately reflect lesion boundaries and color information when compared to the original images. Similarly, in the "DF" class, the baseline model produces unnatural textures, such as the keratinized surface of the lesion. In contrast, our method, utilizing multi-level embeddings, effectively learns and represents boundaries, color, and texture, resulting in more natural and faithful representations.

Fig 6 presents a visual comparison between samples generated by the proposed method and the baseline model, Stable Diffusion. The proposed method more realistically preserves lesion boundary patterns, fine textures, and color distributions compared to the baseline model. This demonstrates the effectiveness of multi-level embeddings in maintaining fine details. While the difference between the two models is less pronounced for classes with abundant training data, the advantages of the proposed method become more apparent for underrepresented classes, such as 'VASC' and 'DF.' Additionally, Fig 7 displays generated samples across all seven classes in the HAM10000 dataset, highlighting the versatility of the proposed approach.

For the quantitative comparison of image results, we measured FID (Fréchet Inception Distance) [78] and IS (Inception Score) [79] as evaluation metrics. FID measures the distribution difference between generated and real data, with lower scores generally indicating higher quality. IS evaluates the diversity and quality of the generated images, where higher scores typically indicate better performance.

Table 2 presents the evaluation metrics for each class along with the average FID and IS scores. The differences in IS scores between the proposed model and the baseline model were not substantial. However, FID scores showed that the proposed method achieved significantly better performance than the baseline. This can be interpreted as the synthetic dataset generated by the proposed method having higher distributional similarity to real data compared to the existing approach. However, it should be noted that FID and IS have known limitations, particularly for non-GAN models like diffusion models. As demonstrated by Ravuri et al. [80], these metrics often fail to predict downstream task performance and may provide

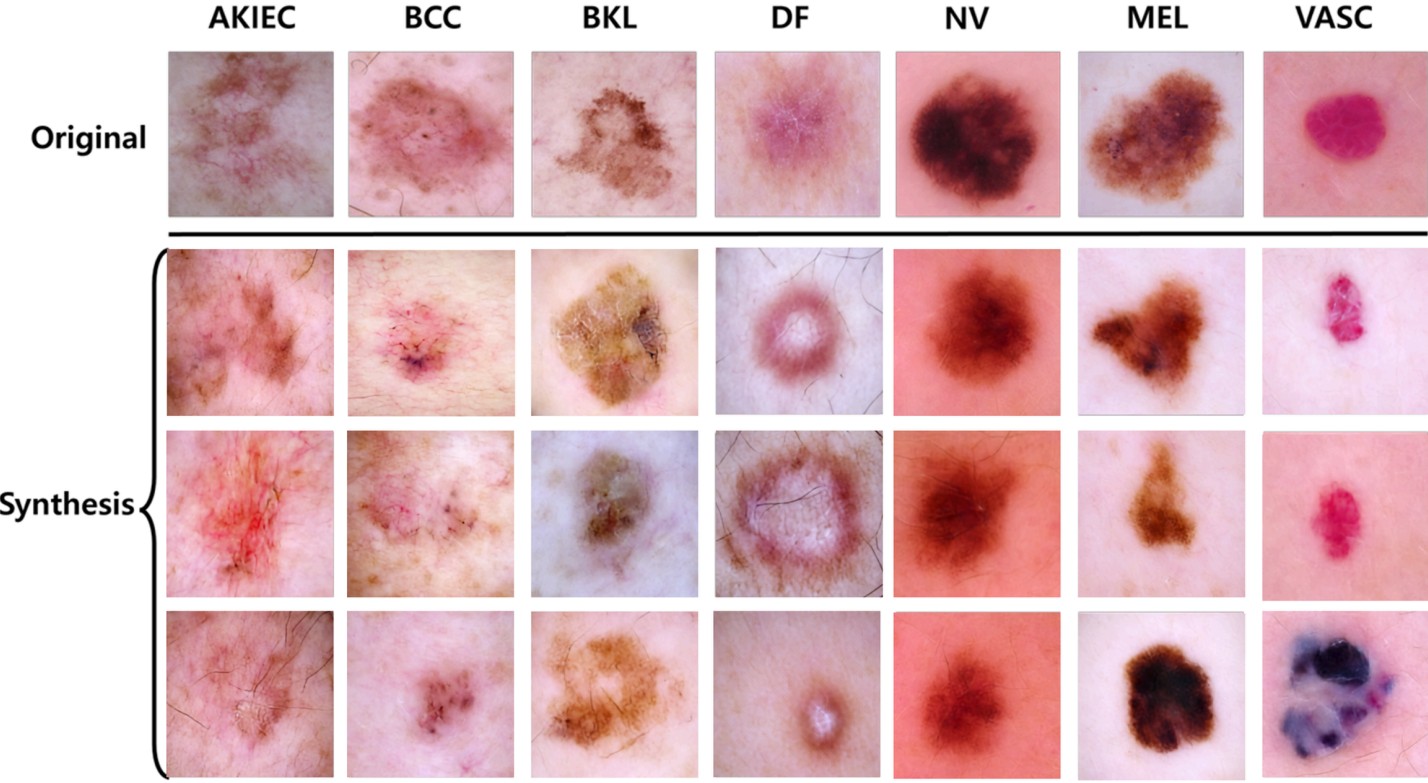

**Fig 7. Generated samples for seven classes.** The synthetic images for each class are visually compared to demonstrate the quality and diversity of the generated data.

**Table 2. Quantitative comparison of image quality between the base model and the proposed method.** This table presents the quantitative results of image quality (FID, IS) for the base model (Stable Diffusion) and the proposed method. While FID scores showed improvement with the proposed method, IS scores showed no significant differences between the models. This indicates that the proposed method demonstrated better performance than the baseline in terms of distributional similarity to real data.

| | FID↓ | IS↑ |
|---|---|---|
| Stable-Diffusion(Base model) | 145.26 | 1.77 |
| Ours Method | **99.21** | **2.26** |

unfair evaluations across different model architectures. Ravuri et al. [80] employed the Classification Accuracy Score (CAS) and demonstrated that FID and IS often fail to correlate with classification performance on real data, with IS sometimes showing misleading negative correlations with actual performance. These limitations are particularly pronounced in medical image synthesis tasks, where learning precise and meaningful data distributions is crucial. While FID and IS are widely used in generative model research, particularly in GAN-based studies, they may not serve as meaningful indicators in this specific context. This underscores the need to develop alternative evaluation metrics that can better assess the relevance and utility of generated data for medical imaging applications.

To complement FID and IS, we assessed whether the synthetic images improve downstream diagnostic models. We evaluated nine different classification models, including relatively simple classifiers such as VGG13 and ResNet18, as well as modern hybrid architectures CNN-based models and Transformer-based architectures like ConvNeXt, Swin Transformer, EVA and CoAtNet. When augmented with synthetic data to ensure sufficient training samples, classification accuracy showed modest improvements compared to using limited real data alone. Detailed results of these experiments can be found in the "Classification downstream task" section below.

## Classification downstream task

The primary objective of this study is to assess the effectiveness of synthetic data in medical applications, particularly for addressing data scarcity challenges in training diagnostic models. To achieve this, we conducted downstream classification experiments using both original and synthetic datasets.

In these experiments, we constructed three dataset configurations to evaluate the impact of synthetic data across different scenarios. We trained nine different classification models spanning two generations of architectures: early-generation CNNs (VGG13, VGG16, VGG19, ResNet18, ResNet34) and modern deep architectures (Swin Transformer, ConvNeXt, EVA, CoAtNet), and compared their classification accuracy.

The composition of each dataset is as follows:

- Original250: 250 original images
- Original500: 500 original images
- Synthetic500: 500 synthetic images
- Mixed1000: original 500 + synthetic 500 images

For some classes in the original 250 and Original 500 datasets, the actual number of Original data samples was insufficient to reach 250 or 500 samples (Actinic keratoses, Dermatofibroma, Vascular lesions). The shortfall was supplemented with basic augmentation techniques such as rotation and flipping that do not affect color or brightness.

Table 3 presents the classification accuracy for each dataset.

The experimental results indicate that, when using the same amount of data, models trained exclusively on synthetic data exhibited slightly lower accuracy compared to those trained on original data. This discrepancy likely arises from the limitation that synthetic data cannot fully capture the complexity of real-world characteristics. However, when combining original and synthetic data to construct a more rich dataset (mixed 1000), the average classification accuracy increased from approximately 80.92% to 84.15%, compared to using original data alone. This suggests that the mixed dataset enhanced model training by improving data diversity.

These results quantitatively demonstrate that while synthetic data may have limitations when used independently, combining it with original data can effectively address challenges associated with data collection. Synthetic data shows particular potential as a valuable resource for mitigating data scarcity in the medical field.

However, the relative benefit from synthetic augmentation varied across architectures. While modern architectures such as Swin Transformer, ConvNeXt, EVA, and CoAtNet generally achieved higher absolute performance than earlier-generation models, certain models like EVA showed a comparatively modest improvement of less than 3% when augmented with synthetic data. This suggests that such architectures may already extract sufficiently rich representations from the original data, leaving relatively less room for improvement solely from synthetic augmentation.

**Cross-dataset Zero-shot Evaluation on PAD-UFES-20.** To further assess the generalization capability of the classifiers trained solely on the HAM10000 dataset, we performed a zero-shot evaluation on the PAD-UFES-20 dataset [81]. As shown in Fig 8, unlike HAM10000 which consists of professional dermoscopic images, PAD-UFES-20 contains clinical photographs captured using smartphone cameras. For a fair comparison, the evaluation was conducted only on the four classes shared between the two datasets: AKIEC, BCC, NV, and MEL.

Nine classification models were evaluated zero-shot, without any fine-tuning on PAD-UFES-20. Table 4 shows that transformer-based architectures such as Swin Transformer (Acc=62.15%) and EVA (Acc=64.27%) achieved the highest accuracy, indicating relatively better robustness to domain shifts compared to CNN-based models. Overall, the accuracy range of 49.89% to 64.27% suggests that generalizing from dermoscopic images to smartphone-based clinical photographs remains challenging, likely due to differences in acquisition devices, lighting conditions, and background context.

## Ablation study

**Ablation on Channel Width and Multi-level Embeddings.** We perform an ablation study across three variants: a 4-channel VAE (4ch), an 8-channel VAE (8ch), and an 8ch

**Table 3. Evaluation of synthetic data effectiveness in downstream classification tasks.** Classification downstream tasks were conducted using four types of data consisting of origin and synthetic data. Training with synthetic images alone resulted in lower classification accuracy compared to original images. However, combining both synthetic and original data to form a larger dataset achieved the highest average classification accuracy.

| | VGG13 | VGG16 | VGG19 | ResNet18 | ResNet34 | Swin-T | ConvNext | EVA | CoAtNet | Average |
|---|---|---|---|---|---|---|---|---|---|---|
| **Origin 250** | 72.71 | 72.86 | 73.57 | 77.14 | 77.86 | 75.71 | 72.86 | 75.86 | 75.00 | 74.84 |
| **Origin 500** | 81.43 | 79.29 | 80.00 | **83.57** | 80.71 | 80.72 | 76.43 | 85.50 | 82.74 | 80.92 |
| **Synthetic 500** | 74.70 | 74.40 | 79.20 | 75.66 | 77.52 | 80.46 | 75.32 | 77.88 | 77.57 | 76.63 |
| **Mixed 1000** | **82.64** | **82.73** | **85.21** | 80.44 | **83.97** | **85.20** | **83.77** | **87.43** | **84.98** | **84.15** |

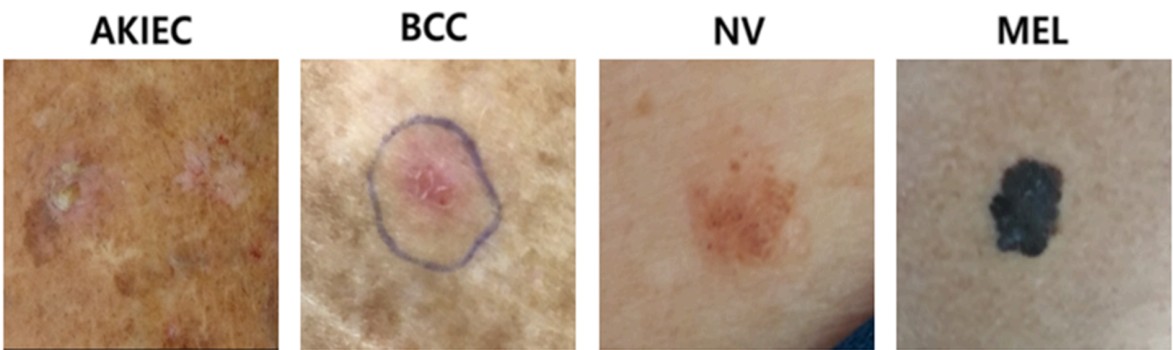

**Fig 8. Example images from the PAD-UFES-20 dataset.** Unlike the HAM10000 dataset, which consists of professional dermoscopic images, PAD-UFES-20 contains clinical photographs captured using smartphone cameras, introducing variations in lighting, resolution, and background.

**Table 4. Cross-dataset zero-shot evaluation results on PAD-UFES-20.** Models were trained on HAM10000 and directly evaluated on zero-shot on PAD-UFES-20 without fine-tuning.

| Model | Accuracy (%) |
|---|---|
| VGG13 | 49.89 |
| VGG16 | 53.39 |
| VGG19 | 58.07 |
| ResNet18 | 51.68 |
| ResNet34 | 59.06 |
| Swin-T | 62.15 |
| ConvNeXt | 61.33 |
| EVA | **64.27** |
| CoAtNet | 58.23 |

model augmented with multi-level CLIP embeddings (8ch+ML). Fig 9 juxtaposes the images generated by each variant, illustrating the progressive improvements.Moving from 4ch to 8ch reduces the excessive bleeding of lesions into the background, suppresses background noise, and reveals fine structures such as capillaries more clearly. This suggests that enlarging the latent dimensionality helps retain high-frequency information.When multi-level embeddings are added, the 8ch+ML model further preserves keratin scales, vascular patterns, and pigment edges with the most realistic texture. The semantic cues provided by the embeddings complement the expanded capacity, enhancing fine-grained depiction.

We quantitatively evaluated the performance of 4ch, 8ch, and 8ch+ML models using FID and IS metrics. As shown in Table 5, both 8ch and 8ch+ML models demonstrate improved FID and IS scores compared to the 4ch baseline. However, when comparing 8ch and 8ch+ML, the 8ch+ML model shows improved FID while exhibiting a decrease in IS scores. This difference indicates that while multi-level embeddings enhance distributional similarity, excessive learning of fine-grained representations may actually reduce diversity. In summary, channel expansion and embedding injection act synergistically to maintain clinically relevant micro-structures while balancing quality and diversity.

**Effect of Class-conditioning.** To further investigate the contribution of the class-conditioning strategy, we compared image synthesis results with and without class-conditioning. For the non-conditioned setting, we completely removed class-conditioning by sampling from the unconditional branch—i.e., no class label embedding was provided— while keeping all other generation settings identical. For the conditional setting, we varied

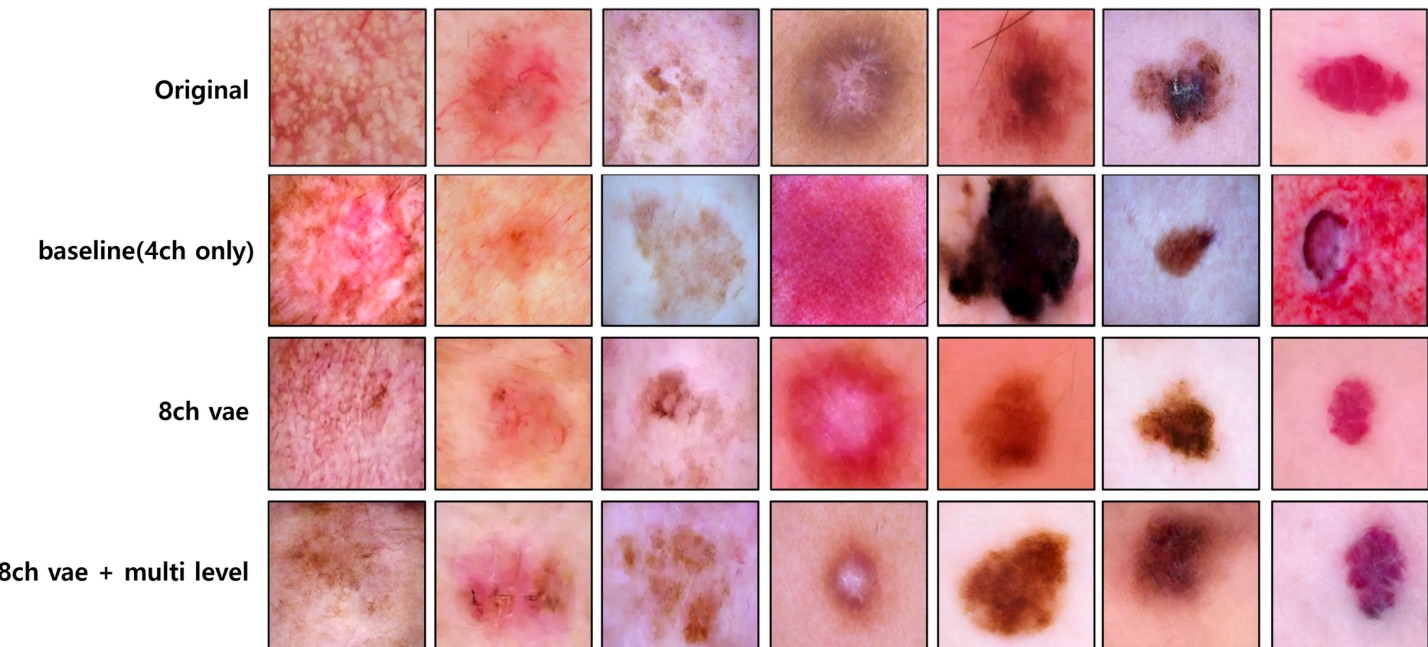

**Fig 9. Visual ablation results.** Visual ablation comparison. Rows, from top to bottom, show (1) the original image, (2) the synthesis produced by the 4-channel VAE (4ch), (3) the synthesis from the 8-channel VAE (8ch), and (4) the synthesis from the 8-channel VAE with multi-level embeddings (8ch+ML). Comparing each column reveals progressively sharper vessel patterns, keratin scales, and overall texture fidelity as channel width is increased and semantic guidance is introduced.

**Table 5. Quantitative ablation study on VAE channel width and multi-level embeddings.** We evaluated three variants using FID and IS metrics: 4-channel VAE (4ch), 8-channel VAE (8ch), and 8ch model with multi-level CLIP embeddings (8ch+ML). Both 8ch and 8ch+ML variants demonstrated improved FID and IS scores compared to 4ch. However, the IS score for 8ch+ML showed a slight decrease compared to the standalone 8ch model, suggesting that excessive learning of fine-grained features through multi-level embeddings may reduce sample diversity.

|  | FID↓ | IS↑ |
|---|---|---|
| 4ch | 145.25 | 1.91 |
| 8ch | 102.21 | **2.64** |
| 8ch+ML | **99.21** | 2.21 |

the classifier-free guidance (CFG) scale among 1, 3, and 5 to observe its effect on the synthesis process. Fig 10 presents class-wise visual comparisons between the conditional and non-conditional generations using identical random seeds.

Without class-conditioning, generated images exhibited weaker class-specific lesion patterns, color distributions, and boundary structures, often resulting in visually similar appearances across different classes. In contrast, applying class-conditioning preserved distinctive features—such as pigment network patterns in MEL, vascular lacunae in VASC—with these discriminative traits becoming more pronounced as the CFG scale increased. This qualitative observation confirms that class-conditioning contributes to maintaining lesion-specific characteristics in our framework.

## Image generation via latent space interpolation for enhanced data diversity

Latent space interpolation was used to naturally sample the transition process from normal to lesion states. By adjusting the interpolation coefficient $\alpha$, defined in Equation (8), from 0.1 to 1.0, images with gradually varying visual characteristics were generated and

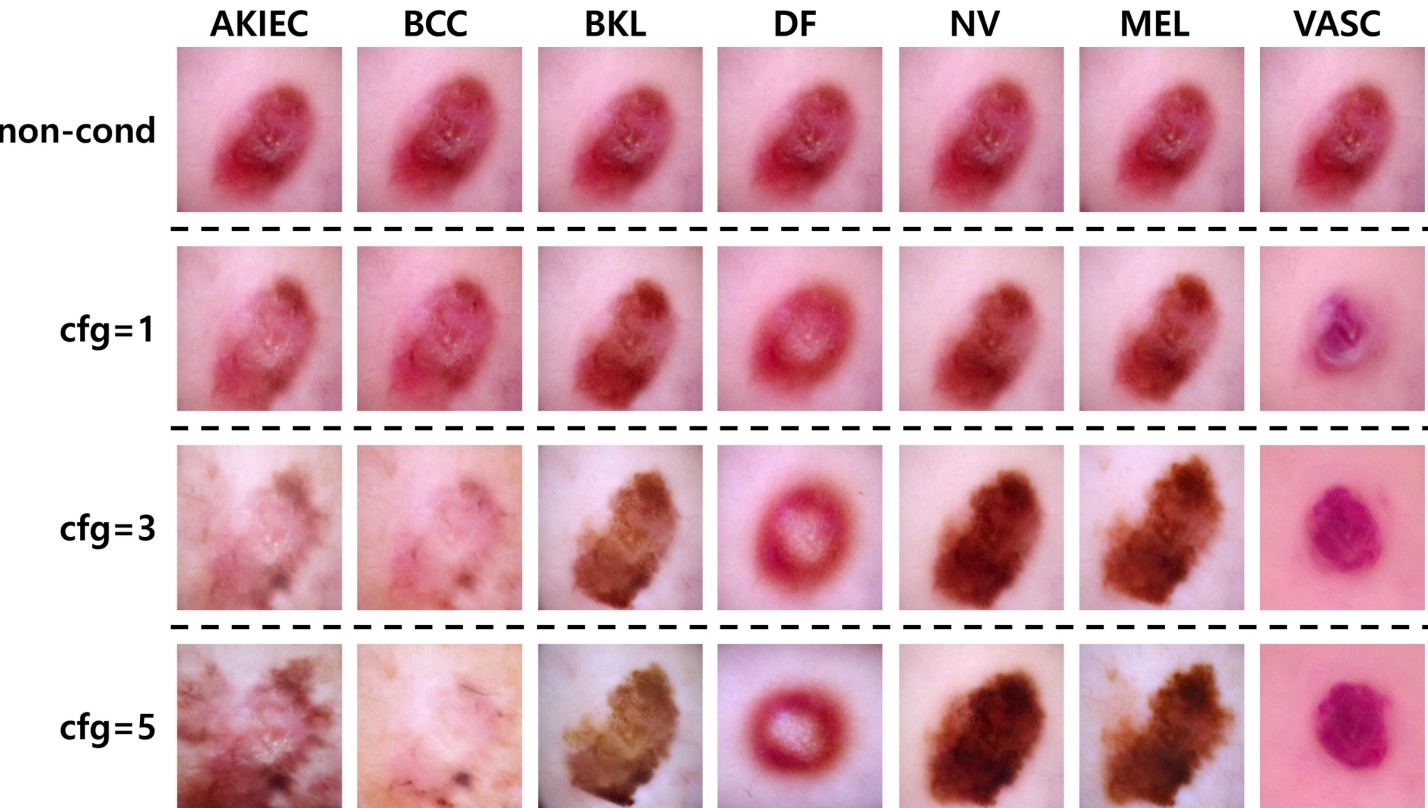

**Fig 10. Effect of class-conditioning and CFG scale on skin lesion synthesis.** Class-wise comparison of non-conditioned generation (top row) and class-conditioned generation at different classifier-free guidance (CFG) scales (1, 3, 5) using identical random seeds. Without class-conditioning (non-cond), images lose distinctive lesion-specific characteristics, resulting in visually similar patterns across classes. Increasing CFG strengthens class-specific morphological features while enhancing overall discriminability between classes.

visualized. As shown in Fig 11, at $\alpha = 0.1$, lesions are almost invisible, while at $0.3 < \alpha < 0.5$, early lesion features gradually begin to appear. At $\alpha = 1.0$, the most prominent lesion features are observed.

This image generation strategy can be utilized as auxiliary data for enhancing the diversity of generated images and may contribute as reference material to support the generalization performance of diagnostic models. It plays a particularly auxiliary role in detecting early lesions and emphasizing the boundary between normal and lesion data.

To evaluate whether the generated images effectively enhance data diversity in the semantic embedding space, we conducted a quantitative analysis based on OpenCLIP. Two types of synthetic image sets were used for this analysis. The first set ($A$) consists of 500 synthetic images randomly generated by the diffusion model. The second set ($A_I$) was constructed by replacing half of the samples in $A$ with images generated via latent space interpolation. For each set, CLIP embeddings were extracted and projected into a two-dimensional space using PCA. Semantic variance was then calculated for each class. Across all classes, the $A_I$ set exhibited higher variance than the $A$ set (Table 6), suggesting that the interpolation-based samples captured a broader range of visual characteristics than randomly generated ones. These findings indicate that the interpolation approach may contribute to expanding the representational space and enhancing the semantic diversity of the training data. The corresponding embedding distributions are visualized in Fig 12.

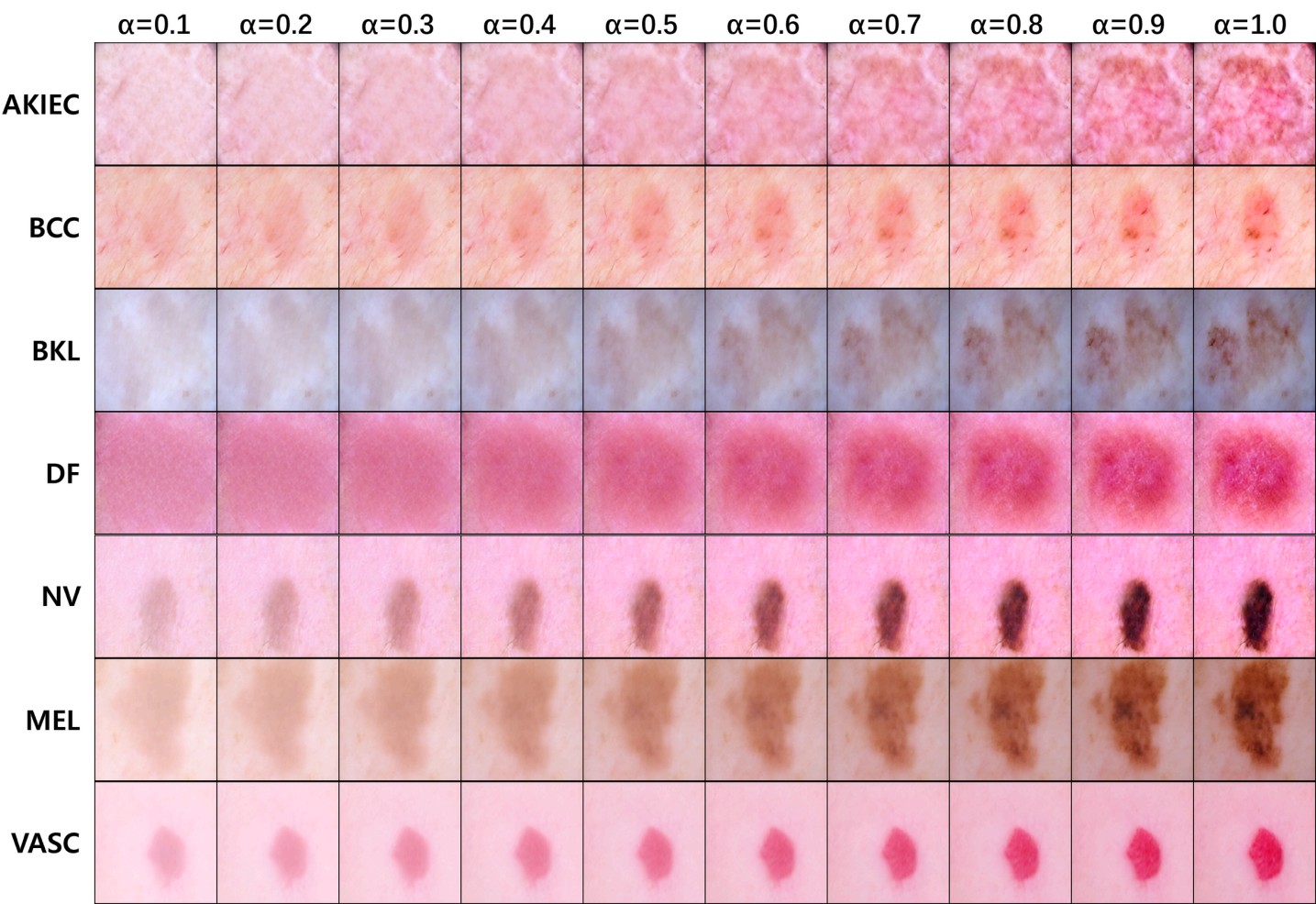

**Fig 11. Comparison of generated samples based on adjustments to the interpolation coefficient $\alpha$.** This figure shows samples generated gradually changing visual characteristics by adjusting the interpolation coefficient $\alpha$ in latent space interpolation. While normal images were generally well generated, completely normal samples could not be achieved when there were significant differences between the normal and lesion regions. Nevertheless, the results demonstrate that adjusting $\alpha$ allows for the generation of diverse reference samples that can enhance training data variability.

**Table 6. Semantic variance comparison between $A$ and $A_I$ using CLIP embeddings and PCA projection.** Across all seven classes, the $A_I$ set—containing partially interpolated samples—consistently exhibited higher variance than the $A$ set, suggesting a potential contribution of interpolation-based samples to enhancing semantic diversity.

| Class | Variance ($A$) | Variance ($A_I$) | Difference ($\Delta$) |
|---|---|---|---|
| AKIEC | 3.7121 | 4.6762 | +0.9642 |
| BCC | 3.0167 | 3.4749 | +0.4582 |
| BKL | 2.8641 | 5.0879 | +2.2238 |
| DF | 2.5864 | 3.8267 | +1.2403 |
| NV | 3.1895 | 6.0547 | +2.8652 |
| MEL | 2.2666 | 4.6148 | +2.3481 |
| VASC | 3.0511 | 4.7766 | +1.7255 |

## Limitations

Synthetic samples did not always generate realistic images. As shown in Fig 13, in some cases, the characteristics of skin lesions were overly emphasized in the wrong direction, resulting

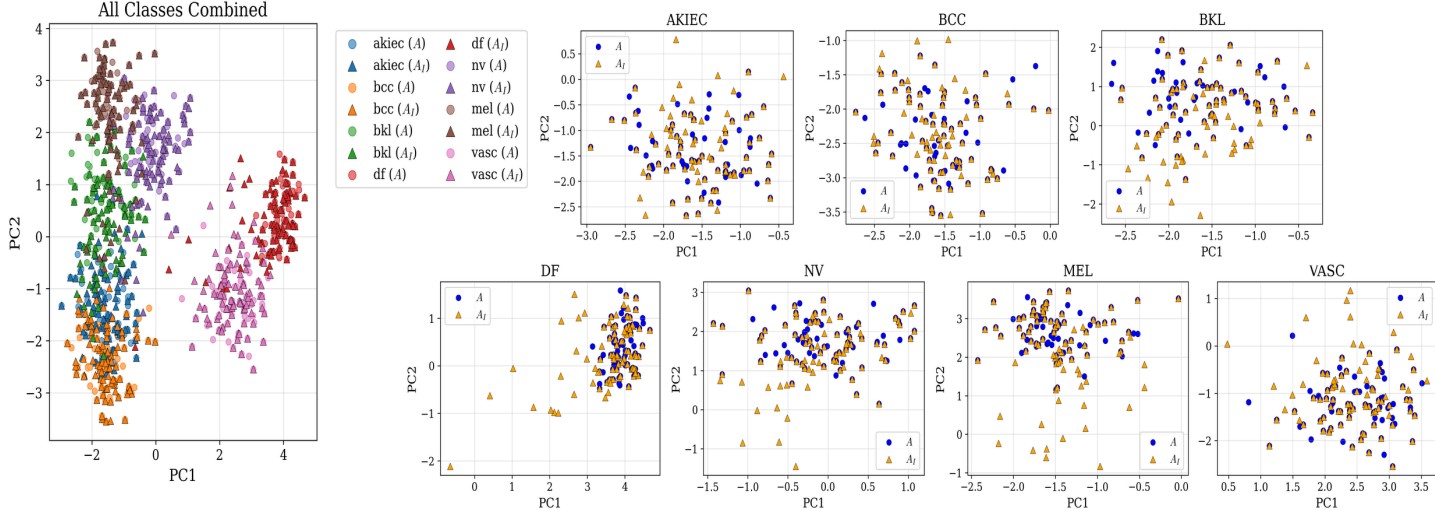

**Fig 12. PCA projection of CLIP embeddings for synthetic image samples A (randomly generated) and $A_I$ (partially interpolated).** The left panel shows all classes combined, while the right panels show each class separately. Triangular markers indicate interpolated samples ($A_I$), and circular markers indicate non-interpolated samples ($A$). In most classes, $A_I$ samples exhibit a wider spatial spread in the embedding space, suggesting a broader coverage of semantic characteristics compared to $A$.

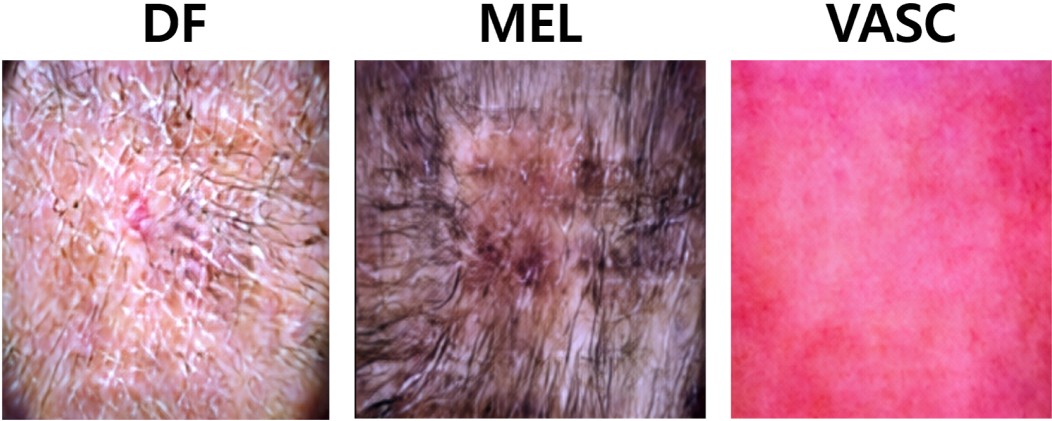

**Fig 13. Failed case samples.** When the influence of fine details, such as hairy regions or color information, is excessively reflected, unrealistic samples are generated.

in suboptimal results. For example, during the process of learning features from the original images, excessive emphasis on areas with hair or specific lesion color information led to such issues.

The current method excels in capturing general visual characteristics such as shape and structure, but when certain features like hair or dominant color information are excessively highlighted, the quality of the generated images can be compromised. These issues underscore the need for balanced feature learning, improved data preprocessing, or complementary techniques to enhance the realism and diversity of synthetic data.

To complement these problems, we found that applying the classical *DullRazor* algorithm [82] to remove hair artefacts before diffusion training can alleviate these issues to some extent. The algorithm detects dark linear structures and inpaints them with neighbouring skin

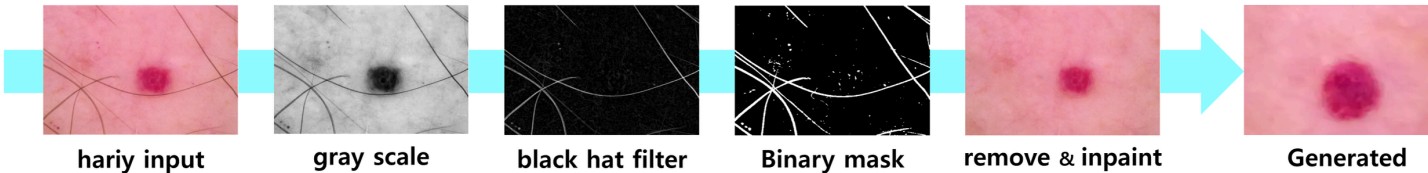

**Fig 14. Hair removal preprocessing using the DullRazor algorithm before image generation.** The DullRazor algorithm systematically removes hair artifacts through a multi-step process: (1) converting the original hairy input image to grayscale, (2) applying a black hat filter to detect and isolate hair structures, (3) creating a binary mask of detected hair regions, and (4) removing hair regions and inpainting the underlying skin texture. This preprocessing step significantly improves the quality of generated images by eliminating hair-related artifacts that could otherwise dominate the synthesis process, resulting in cleaner and more clinically relevant synthetic skin lesion images.

colours, yielding cleaner lesion regions for both real and synthetic images. As shown in Fig 14, the figure demonstrates the hair removal process through the DullRazor algorithm and the synthesis results using this approach.

A second limitation is that our latent interpolation constitutes only a binary lesion interpolation. Although colour and structure intensify as the interpolation coefficient $\alpha$ increases, this visual progression is heuristic in nature may deviate from clinically defined severity, and therefore should not yet be regarded as a strict ordinal scale. We plan to conduct a small proxy experiment using the Derm7pt [83] checklist to assess the monotonic association between $\alpha$ and clinical severity grading, aiming to quantify this relationship using Kendall's $\tau$. Currently, no public dermatology dataset provides ordinal severity labels—existing sets such as HAM10000 and ISIC supply only diagnoses. Furthermore, according to clinical specialists, we note that dermoscopic images in datasets such as HAM10000 may not provide sufficient clinical information for reliable severity grading. In real-world clinical settings, severity assessment typically involves comprehensive evaluation of lesion size, symmetry, spread, ulceration, and morphological context. This limitation indicates that any perceived progression in our interpolation results should be interpreted as an approximate visual continuum rather than a clinically grounded severity concept. Moreover, while our severity interpolation method yields a visually plausible progression, its alignment with clinical grading systems remains unverified.

## Conclusions

In this study, we proposed a novel diffusion-based data augmentation technique to address the challenges of data scarcity in skin disease diagnosis. By enhancing the Stable Diffusion model, we introduced an 8-channel Variational Autoencoder (VAE) to expand latent space capacity and reduce artifacts, thereby preserving fine-grained details crucial for medical imaging tasks. To further improve detail representation, multi-level embeddings extracted using a pre-trained CLIP image encoder were integrated into the diffusion denoising process through adaptive layers.

Our experimental results demonstrate that the proposed method effectively generates high-quality synthetic images, improving the classification accuracy of skin disease models. Specifically, combining synthetic data with real data increased the average classification accuracy from 80.92% to 84.15%, proving the utility of our approach in mitigating data scarcity and enhancing diagnostic performance.

Despite these contributions, Some synthetic images exhibited unnatural artifacts due to the overemphasis of fine details, such as hair regions and dominant color features. Moreover, when generating normal images, significant differences between lesion regions and

the surrounding background sometimes prevented the creation of completely realistic normal images. To address these issues, future work could incorporate more precise segmentation techniques to accurately remove hair regions and leverage larger, high-capacity models specifically tailored for skin-related tasks, thereby improving preprocessing and achieving better feature balance. Additionally, developing more meaningful evaluation metrics tailored for medical image synthesis could help assess the quality and clinical utility of synthetic data more effectively.

In addition, the proposed image generation method—based on segmentation masks, inpainting techniques, and latent space interpolation—still leaves clinical limitations in severity interpretability. The generated results through interpolation do not establish definitive correlations with clinical severity, and thus should be interpreted as heuristic visual progressions intended to increase data diversity rather than as indicators of actual disease severity. Moreover, currently available public data in the skin disease domain lacks severity-related information. To address this limitation, future research will explore collecting skin severity data validated by clinical specialists, quantifying severity information, and applying it to practical learning processes.

In conclusion, this study presented an effective improvement to medical image data augmentation techniques by proposing a data generation method that preserves fine-grained details. As additional study, the proposed latent space interpolation-based approach, while not completely overcoming clinical limitations, can serve as auxiliary material for diverse data for enhancing visual diversity in generated samples. This study contributes to enhancing the robustness and generalization performance of skin disease diagnostic models and holds potential for expansion to other medical imaging domains.

## Author contributions

**Conceptualization:** Mujung Kim, Jisang Yoo, Soonchul Kwon.

**Data curation:** Byung Jun Kim, Changsik John Pak, Chong Hyun Won, Suk-Ho Moon.

**Formal analysis:** Mujung Kim.

**Methodology:** Mujung Kim, Soonchul Kwon.

**Project administration:** Jisang Yoo.

**Software:** Mujung Kim.

**Supervision:** Jisang Yoo.

**Validation:** Mujung Kim, Byung Jun Kim, Woo Jin Song, Han Gyu Cha, Kyung Hee Park.

**Visualization:** Mujung Kim, Soonchul Kwon.

**Writing – original draft:** Mujung Kim.

**Writing – review & editing:** Jisang Yoo, Soonchul Kwon, Byung Jun Kim.

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
