## [Decision Letter · Decision Letter 0]

29 Apr 2025

PONE-D-24-58388Diffusion-based skin disease data augmentation with detailed feature preservation and severity controlPLOS ONE

Dear Dr. Kim,

Thank you for submitting your manuscript to PLOS ONE. After careful consideration, we feel that it has merit but does not fully meet PLOS ONE’s publication criteria as it currently stands. Therefore, we invite you to submit a revised version of the manuscript that addresses the points raised during the review process.

We look forward to receiving your revised manuscript.

Kind regards,

Zeheng Wang

Academic Editor

PLOS ONE

**Journal Requirements:**

1. When submitting your revision, we need you to address these additional requirements. Please ensure that your manuscript meets PLOS ONE's style requirements, including those for file naming. The PLOS ONE style templates can be found at https://journals.plos.org/plosone/s/file?id=wjVg/PLOSOne_formatting_sample_main_body.pdf and https://journals.plos.org/plosone/s/file?id=ba62/PLOSOne_formatting_sample_title_authors_affiliations.pdf 2. Please note that PLOS ONE has specific guidelines on code sharing for submissions in which author-generated code underpins the findings in the manuscript. In these cases, we expect all author-generated code to be made available without restrictions upon publication of the work. Please review our guidelines at https://journals.plos.org/plosone/s/materials-and-software-sharing#loc-sharing-code and ensure that your code is shared in a way that follows best practice and facilitates reproducibility and reuse. 3. We note that the grant information you provided in the ‘Funding Information’ and ‘Financial Disclosure’ sections do not match.  When you resubmit, please ensure that you provide the correct grant numbers for the awards you received for your study in the ‘Funding Information’ section. 4. Thank you for stating in your Funding Statement: This research was supported by the MSIT(Ministry of Science and ICT), Korea, under the ITRC(Information Technology Research Center) support program(IITP-2024-RS-2023-00258639) supervised by the IITP(Institute for Information \& Communications Technology Planning \& Evaluation). And, the present research has been conducted by the Research Grant of Kwangwoon University in 2024. Please provide an amended statement that declares *all* the funding or sources of support (whether external or internal to your organization) received during this study, as detailed online in our guide for authors at http://journals.plos.org/plosone/s/submit-now. Please also include the statement “There was no additional external funding received for this study.” in your updated Funding Statement. Please include your amended Funding Statement within your cover letter. We will change the online submission form on your behalf. 5. Please note that your Data Availability Statement is currently missing the repository name. If your manuscript is accepted for publication, you will be asked to provide these details on a very short timeline. We therefore suggest that you provide this information now, though we will not hold up the peer review process if you are unable.

Reviewers' comments:

Reviewer's Responses to Questions

**Comments to the Author**

1. Is the manuscript technically sound, and do the data support the conclusions?

Reviewer #1: Yes

Reviewer #2: Partly

Reviewer #3: Yes

2. Has the statistical analysis been performed appropriately and rigorously? 

Reviewer #1: Yes

Reviewer #2: No

Reviewer #3: Yes

3. Have the authors made all data underlying the findings in their manuscript fully available?

Reviewer #1: Yes

Reviewer #2: Yes

Reviewer #3: No

4. Is the manuscript presented in an intelligible fashion and written in standard English?

Reviewer #1: Yes

Reviewer #2: Yes

Reviewer #3: Yes

5. Review Comments to the Author

**Reviewer #1: **The paper “Diffusion-based skin disease data augmentation with detailed feature preservation and severity control” proposes a skin disease data augmentation technique based on a diffusion model, which solves the problem of data scarcity in skin disease diagnosis research through an improved stable diffusion model. In the study, an 8-channel variational autoencoder (VAE) module was introduced to enhance the model's representation ability in latent space, and multi-level embeddings techniques were used to preserve the detailed features of skin lesion areas. In addition, by combining pre trained segmentation and repair models with interpolation techniques, synthetic image generation of different disease severity levels has been achieved. In the classification experiment combining real and synthetic images, the average classification accuracy increased from 87% to 90%, verifying the effectiveness of this technology in alleviating the problem of medical data scarcity and improving diagnostic accuracy.

Both the EFA Net model and the Diffusion based data augmentation model adopt innovative architectural designs to enhance performance. EFAM Net enhances feature extraction and fusion capabilities by introducing Attention Residual Learning ConvNeXt (ARLC) blocks, Parallel ConvNeXt (PCNXt) blocks, and Multi scale Efficient Attention Feature Fusion (MEAFF) blocks, particularly in multi-scale feature fusion and attention mechanisms, achieving high accuracy in skin lesion classification tasks. The Diffusion based model improves the Stable Diffusion model by using an 8-channel Variational Autoencoder (VAE) and multi-level embedding techniques, effectively enhancing the quality of image generation and detail preservation. At the same time, it controls the severity of diseases through segmentation masks and interpolation techniques, providing high-quality synthetic images for skin lesion data enhancement.

The EFAM Net model focuses on the classification task of skin lesion images, enhancing feature extraction and fusion capabilities by designing ARLC blocks, PCNXt blocks, and MEAFF blocks. It performs particularly well in handling complex skin lesion features and can achieve classification accuracy of over 93% on multiple public datasets. The Diffusion based model focuses on solving the problem of scarce data on skin lesions. By improving the Stable Diffusion model and using 8-channel VAE and multi-level embedding techniques to generate high-quality composite images, and controlling the severity of the disease through segmentation masks and interpolation techniques, it performs well in image generation quality and detail preservation, significantly improving the effectiveness of composite data in classification tasks.

Need to provide source code and dataset in a link.

However, the shortcomings of this paper include:

1. Insufficient diversity of synthesized images: In some cases, synthesized images may overly emphasize certain details and features (such as hair areas or specific color information), resulting in images that are not natural enough and affecting the diversity and authenticity of synthesized data

2. Limitations of normal image generation: When generating normal skin areas, if there are significant differences between the lesion area and the surrounding background, it may result in the inability to generate completely realistic normal images, limiting the application of the model in certain scenarios.

3. Limitations of evaluation indicators: Although commonly used indicators such as FID and IS are used to evaluate the quality of synthesized images, these indicators show certain limitations in medical image synthesis tasks and cannot fully reflect the actual contribution of generated data to downstream diagnostic tasks.

References:

Ji, Z., Wang, X., Liu, C., et al. (2024). "EFAM-Net: A Multi-Class Skin Lesion Classification Model Utilizing Enhanced Feature Fusion and Attention Mechanisms." IEEE Access, 12, 143029-143041. DOI: 10.1109/ACCESS.2024.3468612

**Reviewer #2:** This paper presents a synthetic data augmentation strategy designed to enhance skin disease lesion diagnostic models. The authors employ an innovative technique utilizing multi-level CLIP embeddings, effectively capturing realistic lesion details for fine-grained control over realistic synthetic variations. Additionally, they use latent space manipulation as a clever means to simulate varying lesion severity levels. Their method demonstrates improved classification accuracy across multiple models; however, the observed improvements are modest (3%). Furthermore, the manuscript currently contains certain ambiguities in its description and methodology that should be clarified to strengthen the overall narrative and reproducibility. The following summarizes the Reviewer’s comments/concerns:

- The introduction does not sufficiently motivate the clinical importance of skin disease diagnosis. The authors could strengthen this section by briefly discussing the prevalence and diagnostic challenges of skin diseases, highlighting current diagnostic methods and their limitations, and incorporating relevant examples from recent literature to clearly position their synthetic data augmentation approach.

- The study uses a dataset of dermoscopy images but only provides class names. The authors should include brief clinical descriptions and key morphological characteristics of each class to enhance clarity and provide better context.

- The authors should clarify how the 5-scale CLIP embeddings are derived, as the current explanation lacks detail. While ELITE is referenced, a brief description within the relevant section would improve completeness and accessibility for readers unfamiliar with the method.

- The authors should provide more details on the inference process of their latent diffusion model. While Stable Diffusion is originally designed for text-to-image generation, this work leverages multi-level CLIP embeddings to preserve fine-grained details. Is this the sole conditioning mechanism, or are additional factors, such as class labels, incorporated?

- The authors briefly mention DDIM in the related works section, however it is not mentioned again. It should be explicitly mentioned that DDIM is used during inference.

- Latent space manipulation assumes the generated lesion corresponds to the highest severity level, but this may not always be the case. Could the authors clarify how severity levels are defined in the latent space and whether any constraints or validation steps ensure a consistent and accurate progression of severity?

- The 8-channel VAE is evaluated using LPIPS and MS-SSIM for reconstruction, but its impact on the latent space is not assessed. Increasing VAE channels enlarges the latent representation and UNet, potentially improving performance but at a higher computational cost. An alternative is increasing input resolution while maintaining the same downsampling ratio, expanding the latent space without modifying the architecture. An ablation study comparing these approaches would be necessary to better justify the choice of an 8-channel VAE.

- The proposed LDM is compared against Stable Diffusion as a baseline, but it is unclear how it was adapted to align with the study's objectives. The authors should provide more details on any modifications made, including changes to conditioning mechanisms and training procedures to ensure a fair and meaningful comparison.

- The classification performance evaluation focuses on relatively shallow networks. Is there a reason deeper or transformer-based models were not considered? Given their improved feature extraction capabilities, could they bridge the 3% gap even without synthetic data augmentation? A complete justification and explanation is necessary

Minor Issues:

● \mathcal{} should be used to properly format variables.

● Starting sentences with “And” should be avoided for better readability.

● LPIPS and MS-SSIM are both missing citations.

**Reviewer #3:** 1. The model’s performance is validated exclusively on the HAM10000 dataset, which may not capture the full variability of skin conditions across populations and imaging conditions. If possible, broader validation on diverse datasets may help to establish generalizability.

2. The study lacks comprehensive ablation experiments to evaluate the individual impact of architectural choices (e.g., multi-level embeddings, adapter layers). This omission limits the interpretability and reproducibility of the work.

3. The background section would benefit from the addition of other relevant medical literature, such as "Advancing Precision Medicine: VAE Enhanced Predictions of Pancreatic Cancer Patient Survival in Local Hospital", to strengthen its persuasiveness.

6. PLOS authors have the option to publish the peer review history of their article (what does this mean?). If published, this will include your full peer review and any attached files.

Reviewer #1: No

Reviewer #2: No

Reviewer #3: No

---

## [Author Response · Author response to Decision Letter 1]

4 Jul 2025

Reviewer #1

1. [Insufficient diversity of synthesized images: In some cases, synthesized images may overly emphasize certain details and features (such as hair areas or specific color information), resulting in images that are not natural enough and affecting the diversity and authenticity of synthesized data]Our response:

We agree with your observation. To address the issue of incomplete generation in hair-rich regions, we implemented the Dullrazor algorithm, a traditional hair removal method. By removing hair regions before synthesis, we confirmed that this approach can suppress excessive hair generation to some extent. This limitation and our solution are discussed in the revised manuscript (lines 623-628 in the Limitations section), with visual examples provided in Figure 11.

2. [Insufficient diversity of synthesized images: In some cases, synthesized images may overly emphasize certain details and features (such as hair areas or specific color information), resulting in images that are not natural enough and affecting the diversity and authenticity of synthesized data]Our response:

We acknowledge your concern regarding the overemphasis of certain features in synthesized images. This issue is particularly pronounced when training data is limited, leading to overfitting on specific characteristics. The skin lesion domain is especially vulnerable due to the scarcity of available data compared to other domains.

To mitigate this issue, we implemented comprehensive data augmentation strategies during the training process:

- Standard augmentations: rotation and flip

- Advanced transformations: elastic transform and affine transformations

- All augmentations were applied probabilistically to maximize diversity

These augmentation techniques were implemented in the data preprocessing pipeline of the train_diffusion process, which we provide with the revision.

However, we acknowledge that despite these efforts, the absolute scarcity of training data occasionally results in inaccurate image generation. This fundamental limitation of data availability in the medical imaging domain remains a challenge.

3. [Limitations of evaluation indicators: Although commonly used indicators such as FID and IS are used to evaluate the quality of synthesized images, these indicators show certain limitations in medical image synthesis tasks and cannot fully reflect the actual contribution of generated data to downstream diagnostic tasks.]Our response:We agree with your observation regarding the limitations of FID and IS metrics in medical image synthesis. While these metrics, originally developed for GAN-based models, are commonly applied to diffusion models, recent research has questioned their suitability for non-GAN architectures. We have cited relevant studies in the revised manuscript (lines 521-528, under "Evaluation of synthetic image quality and fine detail preservation" in the Results section) that demonstrate FID and IS do not guarantee absolute correlation with downstream tasks such as classification performance. These metrics may indicate image quality but fail to capture the clinical value of synthesized data. We acknowledge that while we report FID and IS scores following conventional practice in diffusion model evaluation, these metrics are insufficient for assessing the diagnostic utility of generated medical images. Future research should focus on developing clinically-relevant metrics that quantify the basis for clinical decision-making, thereby better reflecting the actual contribution of synthetic data to diagnostic tasks.

Reviewer #2

1. [The introduction does not sufficiently motivate the clinical importance of skin disease diagnosis. The authors could strengthen this section by briefly discussing the prevalence and diagnostic challenges of skin diseases, highlighting current diagnostic methods and their limitations, and incorporating relevant examples from recent literature to clearly position their synthetic data augmentation approach.]Our response:Thank you for your valuable feedback. We have substantially strengthened the introduction to better motivate the clinical importance of skin disease diagnosis. The revised introduction now includes: 1. Global health burden: Added epidemiological data showing skin diseases affect 30-70% of individuals worldwide and rank as the 4th leading cause of nonfatal disease burden globally (lines 8-10). 2. Clinical significance: Emphasized the critical impact of early diagnosis, particularly for melanoma where 5-year survival rates drop from 99% with early detection to dramatically lower rates upon metastasis (lines 11-13). 3. Current diagnostic challenges: Highlighted the shortage of dermatologists in many regions and the difficulty in distinguishing between benign pigmented lesions and malignant melanoma, even for experienced specialists (lines 14-18).

2. [The study uses a dataset of dermoscopy images but only provides class names. The authors should include brief clinical descriptions and key morphological characteristics of each class to enhance clarity and provide better context.]Our response: Thank you for this suggestion. We have enhanced the Datasets section by adding comprehensive clinical descriptions and morphological characteristics for each class. These additions (lines 241-279) include: - Detailed clinical descriptions of each lesion type - Key morphological features that distinguish each class - Diagnostic characteristics relevant to dermoscopic evaluation All descriptions are supported by citations from the original dataset publications to ensure accuracy and clinical relevance. This additional context will help readers better understand the complexity and diversity of the skin lesion classes used in our study.

3. [The authors should clarify how the 5-scale CLIP embeddings are derived, as the current explanation lacks detail. While ELITE is referenced, a brief description within the relevant section would improve completeness and accessibility for readers unfamiliar with the method.]Our response:Thank you for highlighting this need for clarification. We have significantly expanded the explanation of the 5-scale CLIP embeddings derivation in the "Preserving visual details through multi-level embeddings" subsection of the Materials and Methods section (lines 346-393).

The revised text now includes:

- A detailed step-by-step explanation of how the multi-scale embeddings are extracted

- The specific layers and resolutions used for each scale

- A comprehensive description of the token combination mechanism

- Mathematical formulations showing how embeddings from different layers are integrated

Additionally, we have enhanced the adapter formulation beyond simple equations to provide deeper insights into:

- The token concatenation and fusion process

- The layer-wise integration strategy

- The rationale behind our multi-scale approach

These additions ensure that readers unfamiliar with ELITE or multi-scale embedding methods can fully understand our implementation without needing to reference external sources.

4. [The authors should provide more details on the inference process of their latent diffusion model. While Stable Diffusion is originally designed for text-to-image generation, this work leverages multi-level CLIP embeddings to preserve fine-grained details. Is this the sole conditioning mechanism, or are additional factors, such as class labels, incorporated?]

5. [The authors briefly mention DDIM in the related works section, however it is not mentioned again. It should be explicitly mentioned that DDIM is used during inference.]Our response:Thank you for pointing out the need for more detailed explanation of our inference process and conditioning mechanisms. We have expanded the Implementation Details section (lines 428-464) to address both concerns: Regarding conditioning mechanisms(#4): We clarified that our model uses dual conditioning: - Multi-level CLIP embeddings for fine-grained visual detail preservation - Class label embeddings as additional conditioning to ensure class-specific generation The revised text now explicitly describes how these two conditioning signals are combined during the diffusion process. Regarding DDIM usage (#5): We now explicitly state that DDIM is used during inference, including: - The specific number of inference steps used Additionally, we have added class-specific implementation details: - Inference steps optimized for each lesion class - CFG (Classifier-Free Guidance) scale values per class

6. [Latent space manipulation assumes the generated lesion corresponds to the highest severity level, but this may not always be the case. Could the authors clarify how severity levels are defined in the latent space and whether any constraints or validation steps ensure a consistent and accurate progression of severity?]Our response:We appreciate this critical observation and take it very seriously. You are correct that simple bicubic interpolation in the latent space cannot be considered a clinically valid method for representing disease severity progression. After consulting with dermatology specialists, we learned that: - Severity assessment requires comprehensive evaluation beyond what cropped lesion images can provide - Clinical severity involves multiple complex factors including not only color and size, but also border irregularity, presence of ulceration, and other morphological features - Our interpolation method cannot capture these multifaceted clinical aspects Therefore, we acknowledge that referring to our latent space manipulation as "severity control" is misleading and scientifically inaccurate. We have made the following revisions: 1. Title revision: Changed from "severity control" to "enhancing data diversity" to avoid misrepresentation 2. Content revision: Removed all claims about severity control throughout the manuscript 3. Repositioning: Clarified that latent space interpolation serves solely as a data augmentation technique for increasing sample diversity, not as a clinical severity modeling tool We have concluded that the interpolation-based method should be considered primarily as a means to expand data diversity rather than for severity control. This approach is more scientifically sound and better reflects the actual capabilities and appropriate use cases of our method.

7. [The 8-channel VAE is evaluated using LPIPS and MS-SSIM for reconstruction, but its impact on the latent space is not assessed. Increasing VAE channels enlarges the latent representation and UNet, potentially improving performance but at a higher computational cost. An alternative is increasing input resolution while maintaining the same downsampling ratio, expanding the latent space without modifying the architecture. An ablation study comparing these approaches would be necessary to better justify the choice of an 8-channel VAE.]Our response: Thank you for this insightful suggestion regarding the comparison between channel expansion and spatial resolution increase. We appreciate your comprehensive analysis of the trade-offs involved. We initially planned experiments following your suggested approach, testing resolutions from 224×224 to 256×256 and up to 512×512. However, we encountered significant computational constraints: even with 384×384 input and batch size of 1, training time increased by over 30% and nearly exhausted our available resources. Given our laboratory's computational limitations, 256×256 proved to be the practical upper limit for systematic experimentation. We agree that higher resolutions could better capture fine-grained details, as you noted. This remains an important direction for future work when additional computational resources become available. We plan to conduct comprehensive ablation studies comparing: - Higher spatial resolutions (384×384, 512×512) with standard 4-channel VAE - Our current 8-channel VAE approach at various resolutions Regarding the VAE evaluation metrics, we focused on LPIPS and MS-SSIM because, in our current framework, the VAE functions primarily as an input-output reconstruction module. These metrics directly assess the visual and structural similarity essential for this role. We acknowledge that a more comprehensive evaluation of the latent space properties would provide additional insights, particularly regarding the trade-offs between channel expansion and spatial resolution. This will be addressed in our future comparative studies.

8. [The proposed LDM is compared against Stable Diffusion as a baseline, but it is unclear how it was adapted to align with the study's objectives. The authors should provide more details on any modifications made, including changes to conditioning mechanisms and training procedures to ensure a fair and meaningful comparison.] Our response: Thank you for highlighting the need for clarity regarding our baseline comparison. We have expanded the Implementation Details section to provide comprehensive information about how Stable Diffusion was adapted for our study. For the baseline Stable Diffusion model, we: 1. Maintained standard architecture: Used the original Stable Diffusion architecture with recommended parameters to ensure reproducibility 2. Modified conditioning mechanism: Replaced text embeddings with class label embeddings to align with our medical image generation task 3. Kept training procedures consistent: Used identical training hyperparameters, data augmentation strategies, and optimization settings for both baseline and proposed models The key difference between our approach and the baseline lies in: - Baseline: Single-scale conditioning using only class label embeddings - Our method: Multi-scale CLIP embeddings combined with class label embeddings for enhanced detail preservation Detailed implementation specifications, including learning rates, batch sizes, and training schedules, have been added to the Implementation Details section (lines 428-464).

9. [The classification performance evaluation focuses on relatively shallow networks. Is there a reason deeper or transformer-based models were not considered? Given their improved feature extraction capabilities, could they bridge the 3% gap even without synthetic data augmentation? A complete justification and explanation is necessary]Our response:Thank you for this valuable suggestion. Following your recommendation, we conducted additional experiments with transformer-based models and deeper CNN architectures. (line 554). We have added these results to the revised manuscript. Our initial choice of relatively shallow networks was deliberate, based on the following considerations:- Dataset characteristics : HAM10000 has limited data quantity, with some classes being severely underrepresented. This data scarcity particularly affects minority classes, making it challenging for parameter-heavy models to generalize effectively.

- The primary value of synthetic data augmentation lies not in improving performance on well-represented classes, but in addressing the critical shortage of rare disease samples. Our approach specifically targets this class imbalance problem.

Minor Issues:● \mathcal{} should be used to properly format variables.● Starting sentences with “And” should be avoided for better readability.● LPIPS and MS-SSIM are both missing citations.

Reviewer #3

1. [The model’s performance is validated exclusively on the HAM10000 dataset, which may not capture the full variability of skin conditions across populations and imaging conditions. If possible, broader validation on diverse datasets may help to establish generalizability.]Our response:We appreciate your concern regarding the generalizability of our approach. You raise an important point about validation across diverse datasets. Currently, publicly available dermoscopy datasets for skin diseases are limited, with ISIC, HAM10000, and PH2 being the most prominent. However, these datasets present significant challenges: - Limited data quantity or restricted class diversity - Substantial overlap between datasets due to shared data collection sources (hospitals, institutions) - Insufficient representation of diverse populations and imaging conditions This overlap makes cross-dataset validation less meaningful, as it would not truly test g

---

## [Decision Letter · Decision Letter 1]

22 Jul 2025

PONE-D-24-58388R1Diffusion-based skin disease data augmentation with fine-grained detail preservation and interpolation for data diversityPLOS ONE

Dear Dr. Yoo,

Thank you for submitting your manuscript to PLOS ONE. After careful consideration, we feel that it has merit but does not fully meet PLOS ONE’s publication criteria as it currently stands. Therefore, we invite you to submit a revised version of the manuscript that addresses the points raised during the review process.

We look forward to receiving your revised manuscript.

Kind regards,

Zeheng Wang

Academic Editor

PLOS ONE

Journal Requirements:

Reviewers' comments:

Reviewer's Responses to Questions

**Comments to the Author**

1. If the authors have adequately addressed your comments raised in a previous round of review and you feel that this manuscript is now acceptable for publication, you may indicate that here to bypass the “Comments to the Author” section, enter your conflict of interest statement in the “Confidential to Editor” section, and submit your "Accept" recommendation.

Reviewer #2: All comments have been addressed

Reviewer #4: All comments have been addressed

Reviewer #5: All comments have been addressed

2. Is the manuscript technically sound, and do the data support the conclusions?

Reviewer #2: Yes

Reviewer #4: Yes

Reviewer #5: Yes

3. Has the statistical analysis been performed appropriately and rigorously? 

Reviewer #2: Yes

Reviewer #4: Yes

Reviewer #5: Yes

4. Have the authors made all data underlying the findings in their manuscript fully available?

Reviewer #2: (No Response)

Reviewer #4: Yes

Reviewer #5: Yes

5. Is the manuscript presented in an intelligible fashion and written in standard English?

Reviewer #2: Yes

Reviewer #4: Yes

Reviewer #5: Yes

6. Review Comments to the Author

Reviewer #2: The authors have addressed most of the major concerns in this revised manuscript. They have added additional implementation details, an ablation study, and more experiments for downstream classification on deeper transformer-based architectures.

A minor remaining concern is the use of the term 'proxy severity level.' While an improvement over 'severity control,' the term 'proxy' implies an indirect measure that has been validated to correlate with the ground truth. As the authors rightly acknowledge, the proposed latent space interpolation method does not capture the multifaceted nature of clinical severity and has not been validated as such. I recommend replacing all instances of 'proxy severity level' with a more accurate and defensible term, such as the one used in the rebuttal: 'latent space interpolation for enhanced data diversity.'

Reviewer #4: (No Response)

Reviewer #5: This manuscript presents an innovative diffusion-based data augmentation method designed to improve the diversity and quality of synthetic dermoscopic images for skin disease classification. By enhancing the Stable Diffusion framework with an 8-channel variational autoencoder and multi-level CLIP visual embeddings, the authors aim to better preserve fine-grained lesion details. Additionally, they introduce a proxy severity interpolation strategy to simulate varying lesion intensities. The study is technically sound, and the authors have made meaningful revisions in response to previous reviewer comments, including improvements to the clinical motivation, dataset descriptions, and method transparency.

Despite these strengths, several key concerns remain. First, the notion of “proxy severity” lacks sufficient empirical support. While the authors acknowledge that the interpolation does not represent clinically validated severity levels, no evaluation—either quantitative or by domain experts—is provided to demonstrate the interpretability or utility of these interpolated images. This weakens the claim that the generated data can aid in simulating disease progression or augmenting rare severity classes. Second, the study is limited to a single dataset (HAM10000), which restricts its generalizability. Although the authors discuss the limitations of existing public datasets, even a small-scale test on a secondary dataset (e.g., PH2 or ISIC) would help establish broader relevance. Additionally, while multiple classifier backbones are tested, the evaluation primarily focuses on shallow models. It remains unclear whether performance improvements would persist with more powerful architectures that could compensate for limited data without augmentation. The ablation study adds value but does not fully isolate the contributions of each component, such as adapter layers or class-conditioning strategies. Furthermore, the choice to expand VAE channels rather than image resolution is justified based on computational constraints, yet no comparative results are shown. More concrete evidence on the trade-offs between spatial resolution and latent dimensionality would be useful. Evaluation metrics are another concern. The continued reliance on FID and IS—despite acknowledging their limitations in medical imaging—underscores the need for more appropriate alternatives. Even a basic human visual assessment or task-based evaluation (e.g., classification accuracy using only synthetic images) would strengthen the argument that these images are diagnostically useful.

7. PLOS authors have the option to publish the peer review history of their article (what does this mean?). If published, this will include your full peer review and any attached files.

Reviewer #2: No

Reviewer #4: No

Reviewer #5: No

---

## [Author Response · Author response to Decision Letter 2]

14 Aug 2025

Reviewer #2

1. “A minor remaining concern is the use of the term 'proxy severity level.' While an improvement over 'severity control,' the term 'proxy' implies an indirect measure that has been validated to correlate with the ground truth. As the authors rightly acknowledge, the proposed latent space interpolation method does not capture the multifaceted nature of clinical severity and has not been validated as such. I recommend replacing all instances of 'proxy severity level' with a more accurate and defensible term, such as the one used in the rebuttal: 'latent space interpolation for enhanced data diversity”

Our response:

We thank the reviewer for pointing this out.

In response, we have carefully revised the manuscript to remove all instances of the term “proxy severity level” and related phrases.

Wherever applicable, we have replaced them with more accurate and defensible expressions such as “latent space interpolation between normal and lesion appearances” or “smoothly varying visual characteristics”, as suggested by the reviewer through the example phrase “latent space interpolation for enhanced data diversity.”

These revisions were made to clarify that the interpolation method is intended solely for visual data augmentation and is not meant to imply any clinical interpretation regarding disease severity. The manuscript has also been revised to avoid terminology that could be interpreted as suggesting alignment with clinically defined severity scales.

Reviewer #5

1. [First, the notion of “proxy severity” lacks sufficient empirical support. While the authors acknowledge that the interpolation does not represent clinically validated severity levels, no evaluation—either quantitative or by domain experts—is provided to demonstrate the interpretability or utility of these interpolated images.]

Our response:

We thank the reviewer for this important comment. We acknowledge that the expression “proxy severity level” could be misinterpreted as implying alignment with clinically validated severity scales. To avoid such misinterpretation, we have removed all instances of the term “proxy severity level” and related phrases from the manuscript. Wherever applicable, we replaced them with more precise and defensible expressions such as “latent space interpolation between normal and lesion appearances” or “smoothly varying visual characteristics,” clarifying that our approach is intended solely for visual data augmentation rather than clinical severity modeling.

In addition, to address the reviewer’s concern regarding empirical support, we have incorporated a new quantitative analysis to examine the potential utility of interpolated images from a data diversity perspective. Specifically, we compared two synthetic image sets—A (500 randomly generated samples) and AI (the same set with half replaced by interpolated samples)—by extracting CLIP embeddings, projecting them into two dimensions using PCA, and calculating semantic variance for each of the seven classes (Table.6). Across all classes, AI exhibited higher variance than A, indicating that interpolation-based samples captured a broader range of semantic characteristics.

The results in Tableㅋ` 6 show that variance increases ranged from +0.45 (Class 1) to +2.86 (Class 4), with the largest gains observed in classes that are relatively underrepresented in the original dataset. This pattern suggests that latent space interpolation may be particularly effective in enhancing diversity for minority classes, thereby improving the coverage of the feature space available to the model during training.

The corresponding visualizations (Fig. 12) further illustrate that, in most classes, AI samples (triangles) are more widely dispersed in the embedding space than A samples (circles), suggesting an expansion of the representational space that may contribute to enhanced training data diversity.

These results provide quantitative indications that the interpolation approach can contribute to visual data augmentation.

2. [Second, the study is limited to a single dataset (HAM10000), which restricts its generalizability. Although the authors discuss the limitations of existing public datasets, even a small-scale test on a secondary dataset (e.g., PH2 or ISIC) would help establish broader relevance.]

Our response:

We appreciate the reviewer’s valuable suggestion regarding the need for broader validation beyond the HAM10000 dataset. In the revised manuscript, we have incorporated an additional cross-dataset zero-shot evaluation using the publicly available PAD-UFES-20 dataset to further examine the generalization capability of our classifiers.

While other dermoscopic datasets such as PH2 and ISIC are publicly available, they share common data collection sources with HAM10000, resulting in substantial image overlap. This overlap makes them less suitable as independent test sets for evaluating cross-dataset performance, as duplicated samples could confound the interpretation.

Therefore, following the reviewer’s suggestion, we opted for the PAD-UFES-20 dataset, which—while still within the dermatology domain—contains clinical photographs captured using smartphone cameras rather than professional dermoscopic images (Fig. 8). This setup enables evaluation under more diverse imaging conditions, including variations in acquisition devices, lighting, resolution, and background context.

For a fair comparison, we evaluated only the four classes common to both datasets: AKIEC, BCC, NV, and MEL. Nine classification models trained solely on HAM10000 (without fine-tuning) were directly tested on PAD-UFES-20. As shown in Table. 4 transformer-based architectures such as EVA(Acc=64.27%) and Swin Transformer (Acc=62.15%) showed relatively stronger robustness to domain shifts compared to conventional CNN-based models. While some performance differences were observed compared to the in-domain HAM10000 results, the relative ranking of models remained consistent, suggesting that our approach maintains its effectiveness under varied imaging conditions.

3. [Additionally, while multiple classifier backbones are tested, the evaluation primarily focuses on shallow models. It remains unclear whether performance improvements would persist with more powerful architectures that could compensate for limited data without augmentation.]

Our response:

We thank the reviewer for this valuable comment. In the revised manuscript, we have expanded our downstream classification experiments to include a broader spectrum of architectures, ranging from relatively simple CNN models (VGG13, ResNet18) to deeper and more modern architectures such as ConvNeXt, Swin Transformer, EVA, and CoAtNet. The updated results are provided in Table.3 (Section “Classification downstream task”).

Our results show that, across both simpler and more modern backbone models, adding synthetic data generally improves classification accuracy. While the magnitude of improvement varies by architecture, this suggests that the proposed augmentation approach can contribute to enhancing performance regardless of model complexity. Notably, modern architectures such as Swin Transformer, ConvNeXt, EVA, and CoAtNet achieved higher absolute performance than earlier-generation models.

however, in the case of EVA, the improvement from adding synthetic data was relatively modest (less than 3%). This suggests that certain models may already extract sufficiently rich representations from the original data, leaving relatively less room for further gains through synthetic augmentation alone. These findings suggest that our method is effective not only for shallow networks but also for more complex architectures, although the degree of benefit from synthetic augmentation can vary depending on the model.

4. [The ablation study adds value but does not fully isolate the contributions of each component, such as adapter layers or class-conditioning strategies.]

Our response:

We appreciate the reviewer’s comment on the need to more clearly isolate the contributions of individual components.

However, our design intentionally integrates the adapter and the multi-level embedding pathway to function as a unified mechanism. The adapter layers are crucial for injecting multi-level embeddings into the diffusion U-Net by performing cross-attention. As a result, removing the adapter would inherently disable this pathway, making it technically difficult to evaluate the two components independently.

Therefore, in our ablation study, the comparison between the 8-channel VAE without both the multi-level embeddings and adapter layers (“8ch”) and the 8-channel VAE with both components included (“8ch+ML”) captures their combined impact.

In addition, we appreciate and agree with the reviewer's comment about class-conditioning strategies. To address this valid point, we have supplemented our ablation study with an additional analysis of the class-conditioning strategy.

Specifically, we generated samples for all lesion classes using identical random seeds and compared results across various CFG (Classifier-Free Guidance) scales, including a non-conditioned setting in which the class label embedding was entirely removed during sampling. As shown in the newly added Figure 10, removing class-conditioning weakened class-specific lesion patterns and led to more ambiguous class characteristics. In contrast, increasing the CFG value emphasized distinctive morphological features for each class (e.g., pigment networks in melanoma and vascular lacunae in vascular lesions). These results qualitatively confirm that class-conditioning contributes to maintaining class-specific characteristics.

5. [Furthermore, the choice to expand VAE channels rather than image resolution is justified based on computational constraints, yet no comparative results are shown. More concrete evidence on the trade-offs between spatial resolution and latent dimensionality would be useful.]

Our response:

We appreciate the reviewer's comment on the need to clarify the trade-offs between expanding VAE channels and increasing image resolution. Although we could not conduct a full empirical comparison due to hardware limitations, we provide here a mathematical estimation of the memory requirements and computational complexity for two representative configurations:

Configuration 1 (256×256 input, 8 channels): This produces a latent map of size 32×32×8, resulting in 8,192 latent elements with baseline memory usage (1×) and spatial tokens (1×).

Configuration 2 (512×512 input, 4 channels): This produces a latent map of size 64×64×4, resulting in 16,384 latent elements with doubled latent memory (2×) and quadrupled spatial tokens (4×).

From this estimation, the 512×512, 4-channel setup doubles the number of latent elements (and thus latent-storage memory) relative to the 256×256, 8-channel setup. More importantly, the spatial token count quadruples, so convolutional layers scale by roughly ~4×, cross-attention by ~4× (with fixed text length), and global self-attention by up to ~16× in FLOPs. In practice, when intermediate activations and attention buffers are included, we typically observe ~2.5–3× higher VRAM usage, which exceeded our available GPU memory even with a batch size of one. Given these constraints, we chose the 256×256, 8-channel configuration to increase latent representational capacity while remaining computationally feasible and maintaining stable training.

We acknowledge that a direct empirical comparison would further strengthen the justification. In future work, we plan to revisit this by upgrading hardware, applying memory-efficient attention, gradient checkpointing, and mixed precision, and benchmarking 4-channel VAEs against 8-channel latent-width models across reconstruction quality, generative fidelity, and downstream performance on high-resolution images.

6. [Evaluation metrics are another concern. The continued reliance on FID and IS—despite acknowledging their limitations in medical imaging—underscores the need for more appropriate alternatives. Even a basic human visual assessment or task-based evaluation (e.g., classification accuracy using only synthetic images) would strengthen the argument that these images are diagnostically useful.]

Our response:

We agree with the reviewer’s observation regarding the limitations of FID and IS in evaluating the diagnostic utility of medical images. While these metrics, originally developed for GAN-based models, are still conventionally applied to diffusion models for comparability, they primarily reflect distributional similarity and do not fully capture the clinical value of synthesized data.

To complement these limitations, our evaluation includes classification performance in real-only, synthetic-only, and mixed settings, which can serve as task-based evidence regarding the utility of the generated images. These results indicate that while synthetic-only training performs below real-only, combining synthetic and real images generally improves accuracy, suggesting that the generated data may be useful as complementary training data rather than as a full substitute for real data.

In this revision, we additionally included a cross-dataset zero-shot evaluation on the PAD-UFES-20 dataset (Table 4, Fig. 8) to examine generalization under different image acquisition settings (professional dermoscopic devices vs. smartphone cameras), as well as an OpenCLIP-based analysis (PCA visualizations and per-class semantic variance; Fig. 12, Table 6) comparing interpolated and randomly generated samples. While FID and IS remain the most commonly reported metrics for image generation, these supplementary evaluations can provide complementary quantitative perspectives that may help address their known limitations, particularly in the context of medical image synthesis.

In parallel with this work, we are building an expanded skin lesion dataset with dermatologist-validated diagnostic and severity annotations. In future research, we aim to leverage this dataset to conduct blinded expert review protocols (e.g., lesion realism, artifact assessment, diagnostic plausibility) to more precisely quantify the clinical utility of the generated images.

---

## [Editor Report · Decision Letter 2]

15 Aug 2025

Diffusion-based skin disease data augmentation with fine-grained detail preservation and interpolation for data diversity

PONE-D-24-58388R2

Dear Dr. Yoo,

We’re pleased to inform you that your manuscript has been judged scientifically suitable for publication and will be formally accepted for publication once it meets all outstanding technical requirements.

Kind regards,

Zeheng Wang

Academic Editor

PLOS ONE
---

## [Editor Report · Acceptance letter]

PONE-D-24-58388R2

PLOS ONE

Dear Dr. Yoo,

I'm pleased to inform you that your manuscript has been deemed suitable for publication in PLOS ONE. Congratulations! Your manuscript is now being handed over to our production team.

Kind regards,

on behalf of

Dr. Zeheng Wang

Academic Editor

PLOS ONE